



# Reinterpreting the Budyko Framework

Nathan G. F. Reaver[1,2], David A. Kaplan[2], Harald Klammler[2,3], and James W. Jawitz[4]

[1]Water Institute, University of Florida, Gainesville, Florida, USA.
[2]Engineering School of Sustainable Infrastructure and Environment (ESSIE), University of Florida, Gainesville, Florida, USA.
[3]Department of Geosciences, Federal University of Bahia, Salvador, Bahia, Brazil.
[4]Soil and Water Science Department, University of Florida, Gainesville, Florida, USA.

*Correspondence to*: Nathan G. F. Reaver (nreaver@ufl.edu)

**Abstract.** The Budyko framework posits that a catchment's long-term mean evapotranspiration ($\bar{E}$) is primarily governed by the availabilities of water and energy, represented by long-term mean precipitation ($\bar{P}$) and potential evapotranspiration ($\overline{E_0}$), respectively. This assertion is supported by the distinctive clustering pattern that catchments take in Budyko space. Several semi-empirical, non-parametric curves have been shown to generally represent this clustering pattern but cannot explain deviations from the central tendency. Parametric Budyko equations attempt to generalize the non-parametric framework, through the introduction of a catchment-specific parameter ($n$ or $w$). Prevailing interpretations of Budyko curves suggest that the explicit functional forms represent trajectories through Budyko space for individual catchments undergoing changes in aridity index, $\left(\frac{E_0}{\bar{P}}\right)$, while $n$ and $w$ values represent catchment biophysical features; however, neither of these interpretations arise from the derivation of the Budyko equations. In this study, we re-examine, reinterpret, and test these two key components of the current Budyko framework both theoretically and empirically. In our theoretical test, we use a biophysical model for $\bar{E}$ to demonstrate that $n$ and $w$ values can change without invoking changes in landscape biophysical features and that catchments are not required to follow Budyko curve trajectories. Our empirical test uses data from 728 reference catchments in the United Kingdom and United States to illustrate that catchments rarely follow Budyko curve trajectories and that $n$ and $w$ are not transferable between catchments or across time for individual catchments. This non-transferability implies $n$ and $w$ are proxy variables for $\frac{\bar{E}}{\bar{P}}$, rendering the parametric Budyko equations under-determined and lacking of predictive ability. Finally, we show that the parametric Budyko equations are non-unique, suggesting their physical interpretations are unfounded. Overall, we conclude that, while the shape of Budyko curves generally captures the global behavior of multiple catchments, their specific functional forms are arbitrary and not reflective of the dynamic behavior of individual catchments.





# 1 Introduction

The Budyko framework represents a catchment's long-term mean evapotranspiration ($\bar{E}$) as a function of the aridity index ($\phi$), which is defined as the ratio of mean rainfall depth ($\bar{P}$) to mean potential evapotranspiration ($\overline{E_0}$). Current understanding of the Budyko framework is the result of hydrological research over more than a century. The approach has seen a resurgence within catchment hydrology in recent years, partially due to its simplicity, analytical elegance, and potential for studying and predicting landscape rainfall partitioning under changing climate and land use (Wang et al., 2016a; Mianabadi et al., 2020). Early investigators proposed equations for semi-empirical curves to describe the aggregate behavior of $\bar{E}$ as a function of $\bar{P}$ and $\overline{E_0}$ for large numbers of catchments (Schreiber, 1904; Ol'Dekop, 1911; Budyko, 1974). Since then, efforts to extend the utility of the Budyko framework have both retained and emphasized the concept of explicit curves, leading to the development of parametric Budyko equations. The parameters of these equations are typically referred to as "catchment-specific parameters" and are generally interpreted as representing the influence of all catchment biophysical features, other than $\bar{P}$ and $\overline{E_0}$, on $\bar{E}$ (Wang et al., 2016a). This interpretation has motivated profound efforts to understand the relationship between biophysical features and catchment-specific parameters (Yang et al., 2007; Donohue et al., 2012; Yang et al., 2009; Shao et al., 2012; Li et al., 2013; Xu et al., 2013; Cong et al., 2015; Yang et al., 2016; Zhang et al., 2018; Abatzoglou and Ficklin, 2017; Xing et al., 2018a; Zhao et al., 2020; Ning et al., 2020b; Ning et al., 2020a; Li et al., 2020c; Li et al., 2020b; Zhang et al., 2019b; Ning et al., 2019; Bai et al., 2019). Numerous studies have also focused on determining the sensitivity of rainfall partitioning to climatic and/or land use changes (Roderick and Farquhar, 2011; Wang and Hejazi, 2011; Yang and Yang, 2011; Wang et al., 2016b; Zhou et al., 2016; Shen et al., 2017; Zhang et al., 2016; Yeh and Tsao, 2020; Zhang et al., 2020; Sinha et al., 2020; Ning et al., 2020b; Liu et al., 2020; Li et al., 2020e; Li et al., 2020a; Liu et al., 2019a; Li et al., 2019; Yang et al., 2018; Xing et al., 2018b; Li et al., 2018; Xiangyu et al., 2020), as well as on deriving causal attribution to changes in this partitioning (Wang and Hejazi, 2011; Xing et al., 2018b; Jaramillo et al., 2018; Mo et al., 2018; Sun et al., 2014; Jiang et al., 2015; Liang et al., 2015; Huang et al., 2016; Zhang et al., 2020; Yeh and Tsao, 2020; Xiangyu et al., 2020; Song et al., 2020; Sinha et al., 2020; Li et al., 2020d; Li et al., 2020a; Deng et al., 2020; Zhang et al., 2019a; Young et al., 2019; Xin et al., 2019; Wang et al., 2019; Lv et al., 2019; Liu et al., 2019c; Lee and Yeh, 2019; Kazemi et al., 2019; He et al., 2019c; He et al., 2019b; He et al., 2019a; Wang et al., 2018; Xu et al., 2014).

Despite this widespread application, several doubts have been raised about the robustness of the assumptions and interpretations that underpin this vast and growing literature, particularly with respect to the parametric Budyko equations. For example, Gentine et al. (2012) suggested that the aggregate Budyko curve





behaviour already reflects the interdependence among vegetation, soil, and climate, and therefore, the inclusion of catchment-specific parameter into the Budyko framework is unnecessary. Additionally, Greve et al. (2015) highlighted that the catchment-specific parameter has no *a priori* physical meaning, cannot be estimated for ungauged catchments, and its specific dependence on biophysical features can vary substantially between

catchments. Finally, Sposito (2017a, 2017b) suggested that the interpretation of the catchment-specific parameter as representing biophysical features does not arise from physical reasoning, therefore, identified statistical relationships between the parameter and biophysical features may be spurious and premature.

Given the recent resurgence of the Budyko framework and its importance to catchment hydrology, we build upon these previous critical observations, presenting a retrospective review of the framework's assumptions

and development, with the overarching goals of harmonizing historical and current interpretations as well as understanding their implications. Specifically, we critically reinterpret two key and interrelated components of the current framework: 1) the concept that explicit curves represent trajectories of individual catchments through climate space; and 2) the parametric forms of the Budyko equation themselves. We contend that many current interpretations of these components are unsupported by the underlying framework, potentially leading researchers

to spurious conclusions about catchment hydrology. However, we stress that the aim of this reinterpretation is not to discard the voluminous efforts put forth using current interpretations of the Budyko framework, but rather to recontextualize the conclusions obtained from them.

We first re-examine interpretations of explicit Budyko curves that ascribe physical meaning to the functional form of the curve, thus implying that explicit curve relationships govern catchment evapotranspiration

(e.g., (Wang et al., 2016a; Wang and Hejazi, 2011; Jiang et al., 2015; Liang et al., 2015; Jaramillo et al., 2018; Zhang et al., 2004; Zhang et al., 2018)). This concept is typically articulated through the suggestion that an individual catchment undergoing *only* changes in aridity index will follow an explicit Budyko curve trajectory ("the catchment trajectory conjecture"). Here we examine the support for this conjecture and test it explicitly, the results of which suggest that specific functional forms of explicit Budyko curves do not have intrinsic physical meaning,

but are instead semi-empirical conceptual tools that describe the general aggregate behaviour of multiple catchments—but do not predict the specific behaviour of individual catchments.

Second, we revisit the parametric Budyko equations that are currently interpreted by most authors to represent more generalized forms of the non-parametric Budyko equations (Budyko, 1974), and which can thus be used to separate the effects of changes in the average climate (i.e., changes in aridity index $\phi$) on $\bar{E}$ from the effects

of all other biophysical features (Wang and Hejazi, 2011; Xing et al., 2018b; Jaramillo et al., 2018; Mo et al., 2018; Sun et al., 2014; Jiang et al., 2015; Liang et al., 2015; Huang et al., 2016; Zhang et al., 2020; Yeh and Tsao, 2020;





Xiangyu et al., 2020; Song et al., 2020; Sinha et al., 2020; Li et al., 2020d; Li et al., 2020a; Deng et al., 2020; Zhang et al., 2019a; Young et al., 2019; Xin et al., 2019; Wang et al., 2019; Lv et al., 2019; Liu et al., 2019c; Lee and Yeh, 2019; Kazemi et al., 2019; He et al., 2019c; He et al., 2019b; He et al., 2019a; Wang et al., 2018; Xu et al., 2014). We argue and demonstrate herein that the two widely accepted parametric Budyko equations are non-unique,

meaning they are only two of many possible single-parameter Budyko equations. Importantly, under the catchment trajectory conjecture, the various versions of the parametric Budyko equations are contradictory, which casts doubt on their current interpretations.

        Additionally, while the catchment-specific parameters in the parametric Budyko equations are typically regarded as empirical, "effective" parameters analogous to, for example, Manning's roughness coefficient in open

channel flow or hydraulic conductivity in groundwater flow, we demonstrate that this is not the case, as their values are not transferable between catchments or across time for individual catchments. For an empirical parameter to be transferable, the specific functional form of the mathematical relationship in which it is contained must be empirically valid. In such cases (e.g., Manning's formula and Darcy's Law), the validated functional form contains information about the physics of its respective system, allowing for the empirical parameter to be consistently and

independently related to physical properties of the system (e.g., channel surface roughness for Manning's roughness coefficient and soil pore size for hydraulic conductivity). In these cases, the effective empirical parameters can be estimated *a priori*, allowing their respective empirical relationships to be used for making quantitative predictions under future conditions (e.g., different hydraulic gradients). We explicitly test the empirical validity of the parametric Budyko equations, with results suggesting that the catchment-specific parameter is non-transferrable.

Thus, the value of catchment-specific parameter cannot be determined without first obtaining estimates of $\bar{P}$, $\overline{E_0}$, and most importantly, $\bar{E}$, effectively rendering it a proxy variable for $\frac{\bar{E}}{\bar{P}}$ that has no additional physical meaning and precluding the use of the parametric Budyko equations in predictive pursuits.

        Our reinterpretation is demonstrated theoretically using a stochastic soil moisture model (Porporato et al., 2004), as well as empirically using data from 728 reference catchments in the United Kingdom (UK) and United

States (US). To provide context for these analyses, we first provide a brief background of the Budyko framework, describe its current dominant interpretations in the literature, and recall Budyko's own interpretation of explicit curves.



## 2 Background

### 2.1 Overview of the Budyko hypothesis and equations

In its foundation, the Budyko framework is an expression of the water balance for a catchment. Over long time periods, it is reasonable to assume that positive and negative short-term changes in catchment storage average

to negligibly small values ($\overline{\Delta S} \approx 0$) with respect to the cumulative long-term volumes involved in the water balance. Thus, with $\overline{\Delta S} = 0$, the long-term average precipitation $P$ is partitioned into either evapotranspiration $E$ or discharge $Q$ from the catchment, yielding

$$\bar{P} = \bar{E} + \bar{Q} , \tag{1}$$

where the overbar indicates mean values. Budyko (1974), among others (e.g. (Ol'Dekop, 1911; Thornthwaite,

1948)), recognized that available water ($\bar{P}$) and energy ($\overline{E_0}$) are the primary drivers of long-term average catchment evapotranspiration, and suggested therefore that $\bar{E}$ is a function of $\bar{P}$ and $\overline{E_0}$,

$$\bar{E} = f_0(\bar{P}, \overline{E_0}) . \tag{2}$$

Several explicit functional forms of $f_0$ have been proposed based on their ability to match the central tendency of observed $\bar{E}$ for multiple catchments across a wide range of climates (Ol'Dekop, 1911; Schreiber, 1904; Bagrov,

1953), with Budyko (1974) putting forth,

$$\bar{E} = \bar{P} \sqrt{\left(1 - e^{-\frac{\overline{E_0}}{\bar{P}}}\right) \frac{\overline{E_0}}{\bar{P}} \tanh\left(\frac{\bar{P}}{\overline{E_0}}\right)} . \tag{3}$$

However, Eq. (3) and other forms of Eq. (2) are unable to explain differences in $\bar{E}$ between catchments with identical $\bar{P}$ and $\overline{E_0}$.

Given this limitation, the original Budyko hypothesis has been modified in an attempt to explain deviations

of individual catchments from the explicit Budyko curves by invoking a function that is implicit in $\bar{E}$ (Yang et al., 2008)

$$\bar{E} = f_1(\bar{P}, \overline{E_0}, \bar{E}) , \tag{4}$$

where for a given $\bar{P}$ and $\overline{E_0}$, there may be more than one value of $\bar{E}$ that satisfies Eq. (4). Using the hypothesized relationship given by Eq. (4) and applying constraints for purely water- and energy-limited conditions, it is possible

to analytically derive explicit forms of $f_1$. It has been shown that there are at least two possible analytical solutions





to Eq. (4). The functional form of the first of these solutions was proposed prior (Turc, 1953; Choudhury, 1999; Mezentsev, 1955) to its formal analytical derivation from Eq. (4) by Yang et al. (2008) and is given by

$$\frac{\bar{E}}{\bar{P}} = \frac{\phi}{[1+(\phi)^n]^{\frac{1}{n}}},$$ (5)

where $n$ is a parameter specific to each catchment. With slightly different assumptions about the structure and boundary conditions of $f_1$, a different parametric form of the Budyko equation can also be derived (Fu, 1981; Zhang et al., 2004)

$$\frac{\bar{E}}{\bar{P}} = 1 + \phi - (1 + (\phi)^w)^{\frac{1}{w}},$$ (6)

where $w$ is also a catchment-specific parameter. This equation was also proposed prior to its formal derivation (Tixeront, 1964; Berkaloff and Tixeront, 1958). Equations (5) and (6) express the evaporative index $\left(\frac{\bar{E}}{\bar{P}}\right)$ as a function of the aridity index $\left(\phi = \frac{\bar{E_0}}{\bar{P}}\right)$, however, equivalent forms for both equations exist that express the R-Index $\left(\frac{\bar{E}}{\bar{E_0}}\right)$ (Yao, 1974) as a function of the humidity index $\left(\frac{1}{\phi} = \frac{\bar{P}}{\bar{E_0}}\right)$ (Hulme et al., 1992). We refer to all of these expressions as the "parametric Budyko equations."

Equation (4) has been interpreted as indirectly capturing unknown catchment-specific factors impacting $\bar{E}$, other than $\bar{P}$ and $\bar{E_0}$. The catchment-specific parameters in Eqs. (5) and (6) ($n$ or $w$) arise in part due to the implicit nature of Eq. (4). Two catchments that experience the same $\bar{P}$ and $\bar{E_0}$, but have a different $n$ or $w$, will have different $\bar{E}$. Higher values of $n$ and $w$ correspond to a higher fraction of $\bar{P}$ becoming $\bar{E}$, with $\bar{E}$ approaching $\bar{E_0}$ in energy-limited systems, and $\bar{E}$ approaching $\bar{P}$ in water-limited systems (i.e., as $n$ or $w$ approaches $\infty$). The lower limits of $n$ and $w$ are 0 and 1, respectively. Mathematically, the value of the catchment-specific parameter describes a family of curves in Budyko space.

## 2.2 Current interpretations of explicit Budyko curves and the parametric framework

Most current interpretations of the functional forms of Budyko curves explicitly acknowledge their semi-empirical nature; however, many studies simultaneously ascribe specific physical meaning to the mathematical expressions. This interpretation suggests that the curves represent trajectories within Budyko space that a catchment will follow if its aridity index changes, which supposedly allows one to make predictions about $\bar{E}$ under different climates (e.g., (Roderick and Farquhar, 2011; Wang and Hejazi, 2011; Yang and Yang, 2011; Wang et al., 2016b; Zhou et al., 2016; Shen et al., 2017; Zhang et al., 2016; Milly et al., 2018)). Critically, this interpretation extends





the concept of an explicit curve from its representation of an emergent global behaviour of multiple catchments to the behaviour of individual catchments, implying that the mathematical expressions describing Budyko curves represent fundamental catchment hydrological processes associated with the aridity index. The specific details of these catchment processes are considered to be unknown, but their integrated effects are represented in the functional form of the explicit curves.

Current interpretations of the catchment-specific parameter follow from the application of explicit curves to individual catchment behaviour. Generally, these interpretations can be grouped into four distinct viewpoints: (1) the catchment-specific parameter is an effective empirical parameter related to biophysical features, and it is possible to discern and understand that relationship (e.g., Wang et al. (2016a)); (2) the parameter is related to biophysical features, but it may not be possible to determine an explicit relationship, therefore it should be treated

probabilistically (Gudmundsson et al., 2016; Greve et al., 2015; Singh and Kumar, 2015); (3) the catchment-specific parameter and parametric forms of the Budyko equation contradict the Budyko hypothesis (Sposito, 2017a, b; Gentine et al., 2012); and (4) the parameter is an arbitrary empirical constant that is generated as a part of the solution to Eq. (4), but it has no *a priori* physical meaning (Greve et al., 2015; Sposito, 2017a; Daly et al., 2019a). In particular, the idea that the catchment-specific parameter is an effective empirical parameter related to

biophysical features (i.e., interpretation 1) has been widely embraced by the catchment hydrology community, which has identified and grouped relevant biophysical features into three categories (Donohue et al., 2012; Harman and Troch, 2014): (1) climate variability; (2) catchment physical processes; and (3) vegetation structure and function. While it is generally well acknowledged that certain climatic variables (e.g., rainfall variability or the fraction of precipitation falling as snow) can influence the catchment-specific parameter (e.g., (Roderick and

Farquhar, 2011; Berghuijs and Woods, 2016)), in practice, many studies effectively neglect this, instead focusing primarily on the role of landscape features or vegetation functioning (Wang et al., 2016a; Zhang et al., 2018; Yang et al., 2016; Greve et al., 2015; Xu et al., 2013; Yang et al., 2008; Donohue et al., 2012; Zhang et al., 2004; Liu et al., 2020; Knighton et al., 2020; Gao et al., 2020; Chen et al., 2020; Wu et al., 2019; Qiu et al., 2019; Liu et al., 2019b; Guo et al., 2019).

2019b; Guo et al., 2019).

The widely held interpretations of explicit curves representing individual catchment behaviour and the catchment-specific parameter representing biophysical/landscape features has led to the development of methods to determine the sensitivity of rainfall partitioning to climate and/or landscape changes for individual catchments (Roderick and Farquhar, 2011; Wang and Hejazi, 2011; Yang and Yang, 2011; Wang et al., 2016b; Zhou et al.,

2016; Shen et al., 2017; Zhang et al., 2016; Yeh and Tsao, 2020; Zhang et al., 2020; Sinha et al., 2020; Ning et al., 2020b; Liu et al., 2020; Li et al., 2020e; Li et al., 2020a; Liu et al., 2019a; Li et al., 2019; Yang et al., 2018; Xing





et al., 2018b; Li et al., 2018; Xiangyu et al., 2020) and multiple methods for decomposing anthropogenic and climatic impacts on rainfall partitioning (Wang and Hejazi, 2011; Xing et al., 2018b; Jaramillo et al., 2018; Mo et al., 2018; Sun et al., 2014; Jiang et al., 2015; Liang et al., 2015; Huang et al., 2016; Zhang et al., 2020; Yeh and Tsao, 2020; Xiangyu et al., 2020; Song et al., 2020; Sinha et al., 2020; Li et al., 2020d; Li et al., 2020a; Deng et al., 2020; Zhang et al., 2019a; Young et al., 2019; Xin et al., 2019; Wang et al., 2019; Lv et al., 2019; Liu et al., 2019c; Lee and Yeh, 2019; Kazemi et al., 2019; He et al., 2019c; He et al., 2019b; He et al., 2019a; Wang et al., 2018; Xu et al., 2014). Additionally, these interpretations have led numerous studies to pursue predictive relationships for the catchment-specific parameter based on various biophysical features (Table S1 in the Supplemental Information) (Yang et al., 2007; Donohue et al., 2012; Yang et al., 2009; Shao et al., 2012; Li et al., 2013; Xu et al., 2013; Cong et al., 2015; Yang et al., 2016; Zhang et al., 2018; Abatzoglou and Ficklin, 2017; Xing et al., 2018a; Zhao et al., 2020; Ning et al., 2020b; Ning et al., 2020a; Li et al., 2020c; Li et al., 2020b; Zhang et al., 2019b; Ning et al., 2019; Bai et al., 2019; Ning et al.). However, these relationships are all statistical or derived from curve fitting, which makes it difficult to develop a mechanistic understanding of causal relationships between the catchment-specific parameter and relevant biophysical features. Additionally, the interpretations typically given for such relationships implicitly assume that the functional forms of either Eq. (5) or Eq. (6) a represent physically meaningful relationship between the aridity and evaporative indices, an assumption which has not been empirically validated. An explicit derivation of $n$ or $w$ in terms of biophysical features would substantially improve our understanding of catchment-specific parameter, as has been noted many times (Zhang et al., 2004; Yang et al., 2008; Donohue et al., 2012; Xu et al., 2013; Greve et al., 2015; Wang et al., 2016a; Zhang et al., 2018). Reaver et al. (2020) fulfilled this literature-identified need by analytically inverting both forms of the parametric Budyko equations, i.e., Eq. (5) and (6), resulting in expressions for $n$ and $w$ only in terms of $\bar{P}$, $\overline{E_0}$, and $\bar{E}$. These expressions allow for $n$ and $w$ to be explicitly expressed in terms of biophysical features through the dependence of $\bar{P}$, $\overline{E_0}$, and $\bar{E}$ on those same features.

## 2.3 Budyko's interpretation of explicit curves

It is useful to recall that Budyko (1974) considered explicit curves to be semi-empirical. While the physical basis for explicit curves is dictated by the conservation of mass and energy (i.e., the curves could never cross the water and energy limits in Budyko space) and the fact that the curves must approach the energy and water limits for increasing humidity (i.e., $\phi \to 0$) and aridity (i.e., $\phi \to \infty$), respectively, their empirical nature comes from the choice of functional form as they transition between the energy and water limits. Any functional form that satisfies the previous two conditions and provides a "good" fit to observed data could thus be a reasonable choice. Indeed,





Budyko's own explicit formulation (Eq. (3)) was simply the geometric mean of the Schreiber and Ol'Dekop formulae, which provided a slightly better fit to observed data (Budyko, 1974). These interpretations are suggested by Budyko's own words:

"*The choice of one or another interpolation function for the transition from the first of these conditions to the second is not very important, since, over most of the range of variation in the parameters of the relationship equation, the appropriate relation deviates little from one or the other boundary condition.*" (Budyko, 1974) (p. 325-326)

From this interpretation, it is clear that the explicit curves resulting from the original Budyko hypothesis, while constrained at their limits by fundamental physical laws, are empirical in nature and not derived from catchment hydrologic processes. It should also be noted that the explicit curve relationships were developed to describe the general behaviour of multiple catchments over a wide range of aridity indices. This gives the nonparametric Budyko curves (e.g., Eq. (3)) some predictive power, albeit in a probabilistic sense. Any given
individual catchment would, on average, be expected to fall close to the explicit curves, but in principle could fall anywhere in Budyko space. Predictions of $\bar{E}$ using the original Budyko curves therefore have a quantifiable uncertainty associated with them. Budyko and Zubenok (1961) showed that this mean error was approximately 10%, which has been confirmed more recently (Gentine et al., 2012).

      Given this background, it is important to recognize the difference between applying a semi-empirical curve
to describe the general behaviour of aggregated catchments and using a similar curve to represent the trajectory of an individual catchment undergoing changes in aridity. The original Budyko curve emerges from the ensemble characteristics of many catchments across a range of aridity indices. Suggesting that Budyko curve behaviour applies to the trajectories of individual catchments may be a reasonable conjecture, but it requires either theoretical justification or empirical validation, both of which are currently lacking. In the following sections (Sect. 3.1.1 and
3.1.2), we describe our methods for testing this assumption using both theoretical models and empirical data.



## 3 Methods

### 3.1 Reinterpreting explicit Budyko curves

#### 3.1.1 Theoretically testing for Budyko curve trajectories

To test the catchment trajectory conjecture, we employed the biophysical stochastic soil moisture model
of Porporato et al. (2004). This model, being physically-based, has been used to lend support to Budyko curves and
in developing relationships between $n$ or $w$ and catchment biophysical features (e.g., (Donohue et al., 2012; Zhang
et al., 2018; Cong et al., 2015)). Porporato et al. (2004) developed a model of the equilibrium probability
distribution of the "effective" relative soil moisture under stationary stochastic rainfall in the form of a marked
Poisson process, from which $\bar{E}$ can also be calculated. It is important to note that this model accounts for the
temporal dependence of rainfall but assumes constant potential evaporation. While this limits some of the specific
conclusions that can be drawn from the model, it is adequate for testing the Budyko curve catchment trajectory
conjecture, as the conjecture cannot be valid generally if it is not valid for catchments with time-invariant $E_0$.

We first write the model of Porporato et al. (2004) in a "Budyko-like" form,

$$\frac{\bar{E}}{\bar{P}} = \frac{\overline{E_0}\bar{x}}{\bar{P}} = 1 - \frac{\overline{E_0}}{\lambda(s_I-s_W)\rho Z_r} \frac{\left(\frac{(s_I-s_W)\rho Z_r}{\alpha}\right)^{\frac{\lambda(s_I-s_W)\rho Z_r}{\overline{E_0}}} e^{-\frac{(s_I-s_W)\rho Z_r}{\alpha}}}{\gamma\left(\frac{\lambda(s_I-s_W)\rho Z_r}{\overline{E_0}}, \frac{(s_I-s_W)\rho Z_r}{\alpha}\right)}, \tag{7}$$

where $\bar{x}$ is the mean of effective relative soil moisture $x = \frac{(s-s_W)}{(s_I-s_W)}$, $s$ is the relative soil moisture, $s_W$ is the relative
soil moisture at wilting point, $s_I$ is the well-watered condition threshold relative soil moisture falling between
saturation (i.e., $s = 1$) and relative soil field capacity, $\rho$ is the soil porosity, $Z_r$ is the effective rooting depth, $\alpha$ and
$\lambda$ are the mean rainfall depth and event frequency for marked Poisson process rainfall, and $\gamma(\ ,\ )$ is the lower
incomplete gamma function. The seven parameters ($s_W, s_I, \rho, Z_r, \alpha, \lambda$, and $\overline{E_0}$), can be rewritten in terms of three
effective parameters, defined as $Z_0 = (s_I - s_W)\rho Z_r$, $\psi = \frac{1}{\alpha}$, and $\eta = \frac{\lambda}{\overline{E_0}}$. This simplifies the expression of Eq. (7)
to,

$$\frac{\bar{E}}{\bar{P}} = 1 - \frac{1}{Z_0\eta} \frac{\psi Z_0^{\eta Z_0} e^{-\psi Z_0}}{\gamma(\eta Z_0, \psi Z_0)}, \tag{8}$$

which we refer to as the "Porporato model" hereafter. The four parameters that correspond to landscape properties
($s_W, s_I, \rho$, and $Z_r$) are combined into a single effective parameter, $Z_0$, which represents maximum soil water
storage available for evapotranspiration, while the three parameters corresponding to the climate ($\alpha, \lambda$, and $\overline{E_0}$)





reduce to two effective parameters, $\psi$ and $\eta$, defined above. Equation (8) could be further simplified into only two effective parameters (Porporato et al., 2004; Harman et al., 2011; Doulatyari et al., 2015), however, doing so reduces the conceptual clarity provided by $Z_0$, $\psi$, and $\eta$, which explicitly distinguish climate and landscape parameters.

We tested the catchment trajectory conjecture by varying the model climatic parameters while holding the landscape parameter constant. If the resulting trajectories are not Budyko curves, the conjecture should be rejected. Notably, there are five qualitatively distinct ways that $\psi$ and $\eta$ can be varied to produce trajectories in Budyko space, giving five test cases of the catchment trajectory conjecture: 1) varying $\psi$ alone, which we denote "variable storm size"; 2) varying $\eta$ alone, which we denote "variable storm frequency"; 3) varying $\psi$ less than $\eta$, which we

denote "storm frequency-dominated aridity" (Trenberth, 2011; Fischer et al., 2014); 4) varying $\psi$ more than $\eta$, which we denote "storm size-dominated aridity" (Fischer et al., 2014); and 5) varying $\psi$ equal to $\eta$, which we denote "variable precipitation flashiness". All of these test cases can be expressed through a functional relationship between the two variables, $\eta = \psi^c$, with $c = 0$ for the variable storm size test case, $c \to \infty$ for the variable storm frequency test case, $c = 2$ for the storm frequency dominated aridity test case, $c = \frac{1}{2}$ for the storm size dominated

aridity test case, and $c = 1$ for the variable precipitation flashiness test case. In all test cases, we set $Z_0 = 2\ m$.

**3.1.2 Empirically testing for Budyko curve trajectories**

Our empirical test of the catchment trajectory conjecture involves tracking the actual trajectories of reference catchments in Budyko space over time and quantifying whether they follow Budyko curves. Reference catchments are defined based on long-term stability of land use. Therefore, any changes to precipitation partitioning

over time in reference catchments must be attributed to climatic factors, and the catchment trajectory conjecture predicts that their expected trajectories through Budyko space must be Budyko curves (i.e., those described by Eq. (5) or (6)). This prediction can be tested by comparing actual Budyko space trajectories of reference catchments computed from empirical observations against the expectation from the catchment trajectory conjecture. If the observed reference catchment trajectories are distinct from the expected Budyko curve trajectories, the conjecture

should be rejected.

For a given reference catchment, estimates of $\bar{P}$ and $\overline{E_0}$ were obtained from daily records of $P$ and $E_0$, while estimates of $\bar{E}$ were calculated from the catchment water balance, $\bar{E} = \bar{P} - \bar{Q}$. Since $\bar{P}$, $\overline{E_0}$ and $\bar{E}$ represent temporal averages, and we were also interested in temporal trajectories of those magnitudes, we computed time series of moving averages for each of the three variables. Different "realizations" of the actual trajectories in terms





of $\overline{\frac{E_0}{P}}$ and $\frac{\bar{E}}{P}$ for each catchment were found by applying moving-average window sizes ranging in annual steps from 1 year to the full length of record. For the full length of record in each catchment, the theoretical (or "conjectured") Budyko curve of Eq. (5) was fitted by adjusting the value of $n$.

The conjecture was tested for each reference catchment by statistically comparing all realizations of its

actual trajectories to its theoretical Budyko curve trajectory using the non-parametric sign test (Holander and Wolfe, 1973). This is a distribution-free test for consistent over- or under-estimation between paired observations (see also Supplemental Information Sect. S2). Moreover, we calculated the maximum deviations of the actual trajectories (using the 10-year averaging window) from the expected Budyko curve trajectory for all reference catchments. These values represent the largest magnitudes of climate-induced changes in precipitation partitioning that would

be misinterpreted as land use induced changes when subscribing to the catchment trajectory conjecture. Finally, we estimated the magnitude of the largest errors in evaporative index that occurred when using the well-established non-parametric Budyko curve instead of the parametric form. This was done by calculating the maximum deviations between Eq. (3) and the actual trajectories (10-year averaging window) for all reference catchments.

The catchments used in our empirical test were reference catchments in the UK and US. The 68 UK

catchments selected (Fig. S1a) were from the Catchment Attributes and MEteorology for Large-sample Studies for Great Britain (CAMELS-GB) dataset (Coxon et al., 2020a, b), which also had membership in the UK Benchmark Network (UKBN2) dataset (Harrigan et al., 2018) and had the highest data-quality metric (a benchmark score of 6). UKBN2 reference catchments have been identified as "near-natural" and are intended to be used for the investigation of climate-driven changes in river flow. The CAMELS-GB dataset contains daily time series of $Q$, $P$,

and $E_0$ for each catchment with contiguous record lengths between 12 and 45 years. The 660 US reference catchments selected (Fig. S1b) were from the original CAMELS dataset (Addor et al., 2017; Newman et al., 2015). All catchments with the CAMELS dataset are considered reference catchments, with minimal land use changes or disturbances and minimal human water withdrawals (Newman et al., 2015). Daily times series of $Q$, $P$, $T_{max}$ and $T_{min}$ with contiguous lengths between 20 and 35 years were available for each US reference catchment. Daily $E_0$

time series were computed from the daily $T_{max}$ and $T_{min}$ values using the Hargreaves potential evaporation equation (Hargreaves and Allen, 2003; Lu et al., 2005; Allen et al., 1998).





## 3.2 Reinterpreting the parametric Budyko framework

### 3.2.1 Catchment-specific parameters as proxy variables for the evaporative index

To understand the limitations of the catchment-specific parameters within the parametric Budyko framework, it is illuminating to first review their origin. In the derivations of both forms of the parametric Budyko equations (Eqs. (5) and (6)), $n$ and $w$ arise as arbitrary constants from mathematical necessity rather than being introduced in relation to any physically relevant quantities (Zhang et al., 2004; Yang et al., 2008). Specifically, they arise as "separation constants" that are used when solving partial differential equations by the method of separation of variables. The most basic interpretation of the catchment-specific parameter, therefore, is that it is an arbitrary constant required for the solutions of Eq. (4) to satisfy the boundary conditions (i.e., the water and energy balances) while allowing catchments to have different values of $\bar{E}$ for a given $\bar{P}$ and $\overline{E_0}$. This is contrary to the prevailing interpretation of the catchment-specific parameter as an empirical effective parameter related to biophysical features (Sect. 1 and 2.2). The association of the catchment-specific parameter to biophysical features seems to have first arisen as conjecture that was subsequently bolstered by statistical and curve fitting relationships (Table S1), rather than being motivated by specific physical processes.

Empirical relationships with effective parameters are common and useful in hydrology (e.g., Manning's formula and Darcy's Law). The usefulness of such relationships comes from their transferability either between similar physical systems or within the same system at different times. For example, Darcy's Law states that under certain constraints (i.e., small flow velocities and laminar flow) the flux of water through a porous medium will change linearly with changes in the hydraulic gradient. As long as the flow velocities within the given medium remain small, the slope of the relationship between the hydraulic gradient and flux (i.e., the hydraulic conductivity) will remain constant, meaning its value is transferable across time for that porous medium system. The linear gradient-flux relationship holds for a wide range of different porous media, which allows the slope of the relationship to be independently related to physical properties of the various systems (e.g., pore size distributions (Wang et al., 2017)). Therefore, the hydraulic conductivity can be estimated *a priori* from information independent of the hydraulic gradient and flux, and thus, its value can be consistently transferred between systems with similar properties (i.e., those with similar porous media). For the parametric Budyko equations to be useful empirical relationships analogous to Darcy's Law, the functional forms of Eq. (5) and Eq. (6) must be empirically valid. Specifically, the formulae must be shown to describe how a catchment's evaporative index changes for a given change in the aridity index (i.e., the catchment trajectory conjecture would need to be shown to be valid).



We test the empirical validity of the parametric Budyko framework and the transferability of the catchment-specific parameter with our empirical test of the catchment trajectory conjecture using the 728 UK and US reference catchments (Sect. 3.1.2). This analysis explicitly tests the hypothesis that catchments' evaporative indices follow parametric Budyko curves through Budyko space when undergoing changes in aridity indices. Our test of the

transferability of the parametric Budyko curves is directly analogous to testing the linear gradient-flux relationship for Darcy's Law.

### 3.2.2 Non-uniqueness of the parametric Budyko equations

Equations (5) and (6) are the most frequently used parametric "Budyko-like" equations and are generally considered the only valid parametric Budyko equations, since they alone satisfy the uniqueness requirement, that

every point within Budyko space belongs to only one curve as defined by the catchment-specific parameters (Zhou et al., 2015; Sposito, 2017a). Other proposed equations either do not satisfy conservation of mass and energy in all cases (e.g., (Zhang et al., 2001)) or do not generally cover all of Budyko space (e.g., (Wang and Tang, 2014)). However, even though Eq. (5) and (6) satisfy these requirements, they have not been shown to be the only valid parametric equations.

Both Eq. (5) and (6) describe a family of curves in a single-parameter (i.e., $n$ or $w$), which (1) asymptotically approach the energy and water limits as the parameter approaches infinity, (2) asymptotically approach zero as the parameter approaches its lower bound, (3) whose value and first derivative approach 0 and 1, respectively, in the humid limit (i.e., $\phi \to 0$), and (4) whose value and first derivative approach 1 and 0, respectively, in the arid limit (i.e., $\phi \to \infty$). Any other single-parameter equation that has these properties could be

an equally valid parametric "Budyko equation". In this sense, neither Eqs. (5) and (6) nor any other possible single-parameter Budyko equation has a particular claim of being the "correct" equation for representing Budyko space.

Here we introduce two additional relationships that conform to all of the properties of parametric Budyko equations (developed fully in Sect. S3 in the Supplemental Information):

$$\frac{\bar{E}}{\bar{P}} = 1 - \left[\frac{\gamma\left(q_n, \frac{q_n}{\phi}\right)}{\Gamma(q_n)}\right] + \left[\frac{\gamma\left(q_n+1, \frac{q_n}{\phi}\right)}{\Gamma(q_n+1)}\phi\right], \tag{9}$$

and

$$\frac{\bar{E}}{\bar{P}} = 1 - \left[\frac{\gamma\left(q_w-1, \frac{\Gamma\left(q_w-\frac{1}{2}\right)}{\phi^2\Gamma(q_w-1)}\right)}{\Gamma(q_w-1)}\right] + \left[\frac{\gamma\left(q_w-\frac{1}{2}, \frac{\Gamma\left(q_w-\frac{1}{2}\right)}{\phi^2\Gamma(q_w-1)}\right)}{\Gamma\left(q_w-\frac{1}{2}\right)}\phi\right], \tag{10}$$



where $q_n$ and $q_w$ are the catchment-specific parameters and $\Gamma(\ )$ is the gamma function. The parameter $q_n$ is analogous to $n$ of Eq. (5), taking values ranging between 0 and $\infty$, and $q_w$ is analogous to $w$ in Eq. (6), taking values ranging between 1 and $\infty$.

## 4 Results and Discussion

### 4.1 Reinterpreting explicit Budyko curves

#### 4.1.1 Theoretically testing for Budyko curve trajectories

The theoretical test of the catchment trajectory conjecture for cases 1 (variable storm size) and 2 (variable storm frequency) generally resemble Budyko curves in that they are monotonically increasing, concave down, and approach the energy and water limits as $\phi$ approaches 0 and $\infty$, respectively (Fig. 1a). While the trajectories of these two cases appear "Budyko-like", they have non-identical shapes (i.e., they follow two distinct paths), contrary to what would be expected from the catchment trajectory conjecture. For test cases 3 (storm frequency-dominated aridity) and 4 (storm size-dominated aridity), neither theoretical catchment trajectory can be described as a "Budyko-like" curve (Fig. 1b) for presenting changes in concavity. When using the relationship $\eta = \psi^2$ (storm frequency-dominated aridity), the trajectory is not even monotonically increasing and actually moves *away* from the water limit with increasing aridity. The main conclusion of this theoretical test is that a catchment undergoing only changes in aridity does not have to follow a Budyko curve, contrary to the the catchment trajectory conjecture.

The Budyko space trajectory for test case 5 (variable precipitation flashiness) is a vertical line at $\phi = 1$, with $\frac{\bar{E}}{P} \to 0$ as $\psi \to 0$ and $\frac{\bar{E}}{P} \to 1$ as $\psi \to \infty$ (Fig. 1c), which is clearly not a "Budyko-like" curve. Additionally, this trajectory shows that the catchment-specific parameter is not independent of climate, and that changes in climate alone (i.e., changing only $\psi$ and $\eta$ in the Porporato model) can result in arbitrary values of the catchment-specific parameters. The main conclusion to be taken from this test is that the catchment-specific parameter can be highly dependent on climate, as is acknowledged in current interpretations of $n$ and $w$ (Sect. 2.2), but is contrary to how the catchment-specific parameter is typically used in practice, namely as purely representative of landscape features alone. In combination, our theoretical tests illustrate that catchments undergoing changes in climate alone *can* follow Budyko-like curves but are not *required* to do so.



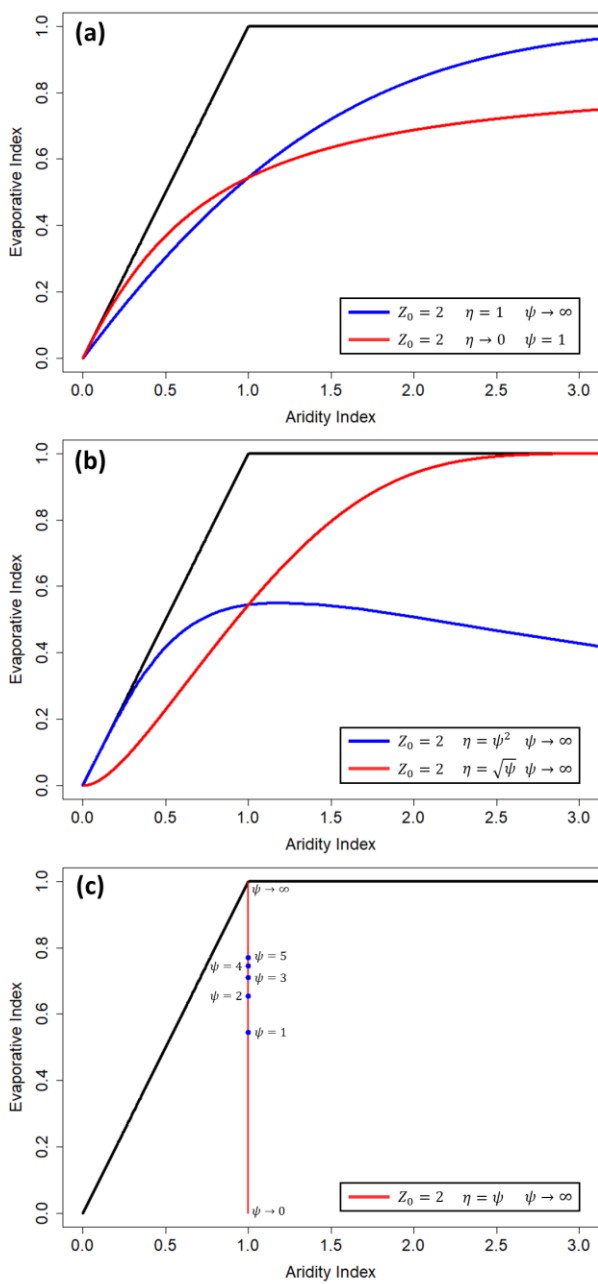

**Figure 1: Resulting trajectories of the theoretical test of the Budyko curve conjecture plotted in Budyko space. The energy and water limits of Budyko space are given as solid black lines. (a) Trajectories for test case 1, variable storm size (blue), and test case 2, variable storm frequency (red). (b) Trajectories for test case 3, storm frequency-dominated aridity (blue), and test case 4, storm size-dominated aridity (red). (c) Trajectory for test case 5, variable precipitation flashiness (red), with locations corresponding to specific values of $\psi$ (blue).**





### 4.1.2 Empirically testing for Budyko curve trajectories

The empirical test of the Budyko curve catchment trajectory conjecture evaluated whether real-world reference catchments, which are not subjected to significant land use change, actually follow Budyko curve

trajectories over time. The catchments investigated span a wide range of aridity indices, climate zones, latitudes, longitudes, and vegetation types, and their global behaviour is in agreement with the non-parametric Budyko curve (Fig. 2). However, individual catchments do not generally follow parametric Budyko curve trajectories (Fig. 2 and Fig. 3a), implying significant errors in the prediction of precipitation partitioning sensitivity based on the catchment trajectory conjecture (Fig. 3b). In addition to the theoretical test of the conjecture (Sect. 3.1.1 and 4.1.1), the

qualitative and quantitative results of the empirical test provide further evidence against the conjecture.

The data for the 728 UK and US reference catchments are shown in Budyko space with their corresponding expected and actual Budyko curve trajectories in Fig. 2. The data generally cluster in a manner reflective of the well-known non-parametric Budyko curve behaviour (blue solid curve). Additionally, the aggregate behaviour of the actual trajectories (red solid curves) also generally follows the non-parametric Budyko curve. However, there

are significant discrepancies between the shape of the overall ensemble cloud of catchments and their actual trajectories versus the corresponding conjectured trajectories (gray dashed curves) for most individual catchments. Many of the curves that would be expected based on the catchment trajectory conjecture span regions of "unpopulated" Budyko space where actual catchments are rarely observed.

Non-parametric sign tests showed that none of the reference catchments consistently followed the Budyko

curves that would be expected based on the catchment trajectory conjecture (i.e., for multiple realizations of actual trajectories using different averaging window sizes). From the total of 24,501 actual trajectory realizations, 23,231 (95%) were found to have consistent differences (p-value < 0.05) from their expected trajectories (i.e., they did not follow Budyko curves), while only 1270 (5%) were found to be statistically indistinguishable (p-value > 0.05). We note that the 5% of actual trajectory realizations for which Budyko curve trajectories could not be rejected is

consistent with the expected 5% that would be accepted due to random chance at a significance level of 95%.

Figure 3a gives examples of actual trajectory realizations (10-year average) that are statistically distinguishable (red curves) and indistinguishable (blue curves) from their expected trajectories (black dotted curves). The maximum deviation between the actual evaporative index (10-year average) and those determined from expected trajectories shown in Fig. 3a is 0.14, corresponding to an absolute relative error of 212%. Figure 3b

gives a histogram of the maximum absolute relative errors in evaporative index between the 10-year average actual





trajectory realizations and expected trajectories for all 728 reference catchments, truncated to a maximum value of 225%. The locations of the errors associated with the example trajectories in Fig. 3a are given by arrows in Fig. 3b, with their colors (red or blue) corresponding to the trajectory's statistical distinguishability. The full range of evaporative index errors spanned from 0.4% to 1991%, with an average value of 26%. This average relative error

for the parametric Budyko framework (26%) is actually larger than that for Eq. (3) (23%), which suggests that the non-parametric Budyko curve is in better agreement with the global behaviour of catchments than the ensemble of parametric curves specifically fit to the individual catchments.

From these results, we can conclude that individual catchments do not generally or consistently follow Budyko curve trajectories as posited by the catchment trajectory conjecture, As such, the use of this conjecture in

hydrological analyses (e.g., precipitation partitioning sensitivity and causal attribution to anthropogenic and climatic impacts) will likely introduce significant errors and may lead to spurious conclusions.

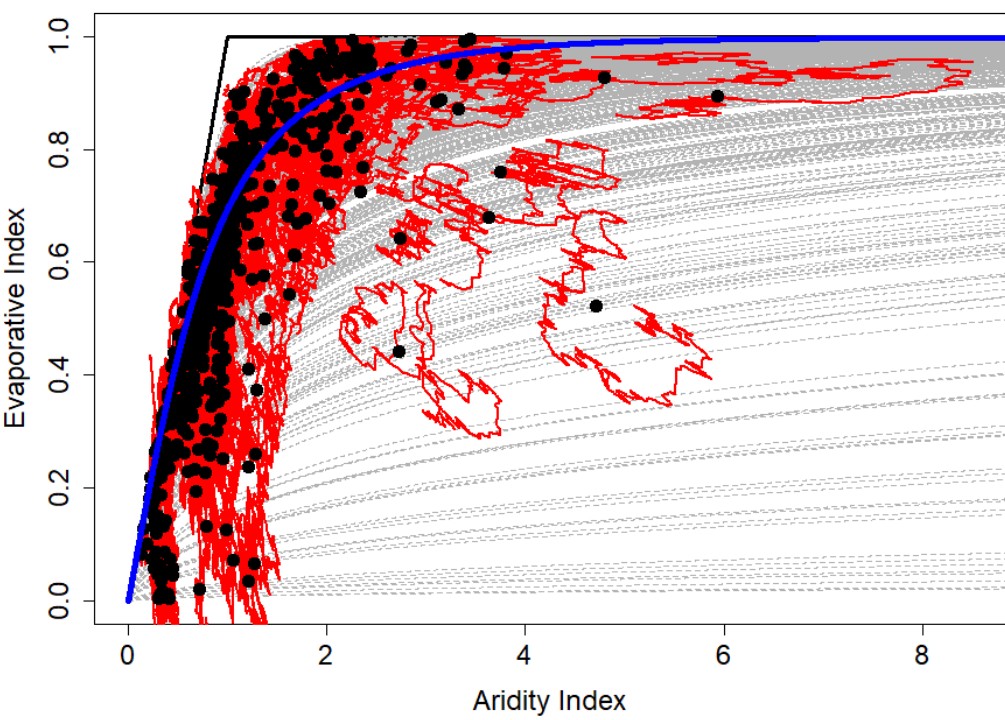

**Figure 2: Budyko space locations (black dots) of the 728 UK and US reference catchments and their corresponding expected Budyko curve trajectories, Eq. (5) (gray dashed curves) and 10-year average actual**

**trajectory realizations (red solid curves). The global behavior of the catchments and their actual trajectories generally agrees with the the non-parametric Budyko, Eq. (3) (blue solid curve) but not the expected parametric Budyko curve trajectories.**





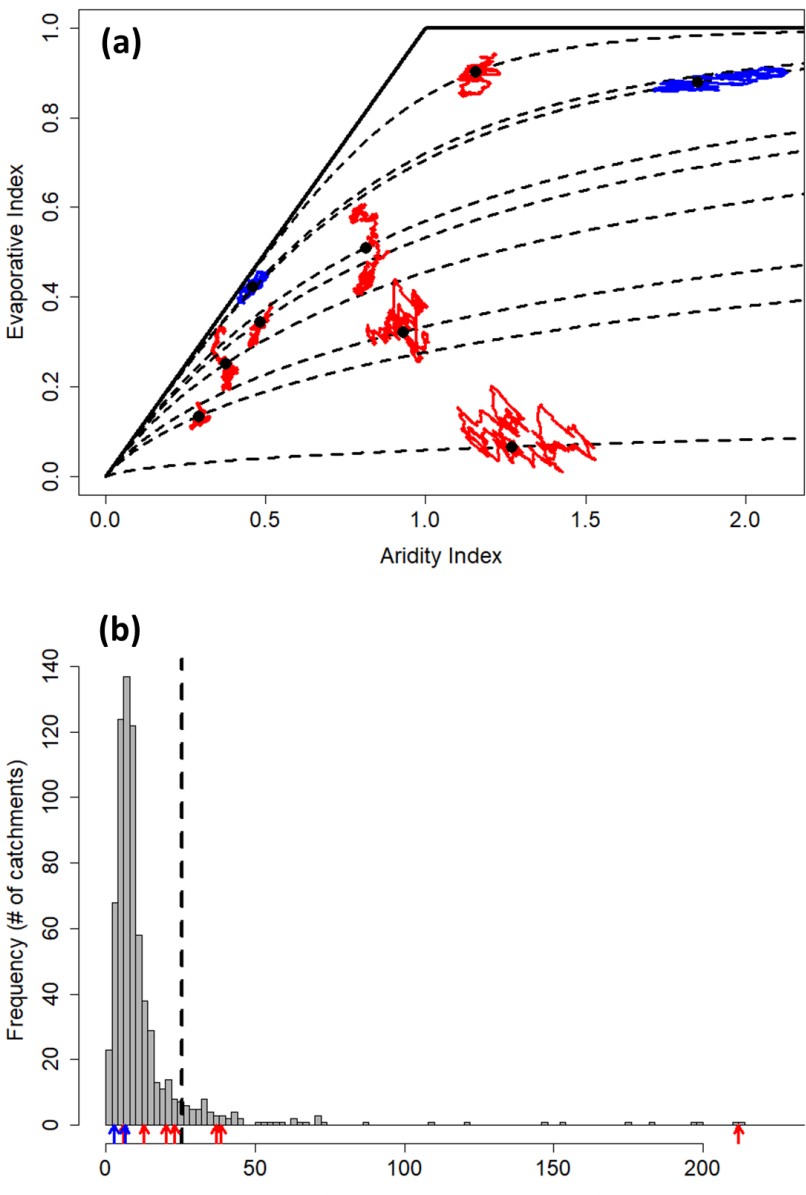

**Figure 3: Comparison of actual catchment trajectories with their corresponding expected Budyko curves,**
5    **Eq. (5), suggested by the catchment trajectory conjecture. (a) Example catchments with statistically**
**distinguishable (red solid curves) and indistinguishable (blue solid curves) actual (10-year average) and**





**expected trajectories (black dashed curves). Catchments, and p-values from the non-parametric sign test, in order of increasing evaporative index: Bull Lake Creek, Wind River Reservation, WY, US, USGS Station 06224000, p ~ 5.84 x 10^-268; River Lune, Killington, Cumbria, UK, NRFA Station 72005, p ~ 6.66 x 10^-16; River Dee, Woodend, Aberdeenshire, UK, NRFA Station 12001, p ~ 6.66 x 10^-16; Shell Creek, Bighorn National Forest, WY, USGS Station 06278300, p ~ 0; River Ithon, Disserth, Powys, UK, NRFA Station 55016, p ~ 4.44 x 10^-16; River Fal, Tregony, Cornwall, UK, NRFA Station 48003, p ~ 0.28; Kiamichi River, Big Cedar, OK, US, USGS Station 07335700, p ~ 8.88 x 10^-16; West Clear Creek, Camp Verde, AZ, US, USGS Station 09505800, p ~ 0.061; Little Withlacoochee River, Rerdell, FL, US, USGS Station 02312200, p ~ 6.66 x 10^-16. (b) Histogram of the in evaporative index maximum absolute relative error between the 10-year average actual trajectory realizations and expected trajectories for all 728 reference catchments, truncated to 225%. Values for the nine example catchments from Fig. (3a) are indicated with arrows colored in correspondence with their statistical distinguishability. The overall mean of the distribution (26%) is given as a vertical black dashed line.**

**4.2 Reinterpreting the parametric Budyko framework**

**4.2.1 Catchment-specific parameters as proxy variables for the evaporative index**

The results of our empirical test of the Budyko curve catchment trajectory conjecture (Sect. 3.1.2 and 4.1.2) strongly suggest that the parametric Budyko equations do not describe the long-term evaporative behavior of individual catchments (i.e., they are not empirically valid). This further suggests that their specific functional forms are not physically meaningful, and the catchment-specific parameter cannot be independently related to physical properties. Thus, $n$ and $w$ are not transferrable either between catchments or between different time points in a single catchment. Without empirical tethers to physical reality, the functional forms of Eq. (5) and (6) do not carry information independent of $\bar{P}$, $\overline{E_0}$, and $\bar{E}$, leaving the parametric Budyko equations under-determined and thus establishing the catchment-specific parameter and $\frac{\bar{E}}{\bar{P}}$ as proxies for each other.

Due to this non-transferability and proxy relationship, it is not possible to solve for $n$ or $w$ without first obtaining $\frac{\bar{E}}{\bar{P}}$, making it impossible to obtain the value of $n$ and $w$ for a catchment *a priori*. When $\bar{P}$ and $\overline{E_0}$ are known, the values of $n$ or $w$ and $\frac{\bar{E}}{\bar{P}}$ are inextricably intertwined since they can only be "measured" by inverting Eq. (5) and (6) using the same evaporative fluxes that are to be eventually estimated. This reliance makes their use in predictive pursuits purely circular. Thus, the parametric Budyko equations are under-determined, as each equation will always contain two unknowns (i.e., $\frac{\bar{E}}{\bar{P}}$ and either $n$ or $w$). This means that for any given $\phi$, there are infinitely many valid combinations of the catchment-specific parameter and $\frac{\bar{E}}{\bar{P}}$ (Sect. S4 and Fig. S2).





The inability to estimate $n$ and $w$ without $\frac{\bar{E}}{\bar{P}}$ has also been noted previously in the literature (e.g., (Zhang et al., 2004;Greve et al., 2015)). This fact is made even more evident by examining the processes used to develop the proposed $n$ and $w$ relationships summarized in Table S1. In every case, first, $\frac{\bar{E}}{\bar{P}}$ is estimated empirically from discharge and rainfall data or from a biophysical model prior to being used to calculate a value for $n$ or $w$, which

is subsequently used to develop the statistical curve fitting relationships. The apparent dependence of the catchment-specific parameter on biophysical features is this directly derived from the dependence of $\frac{\bar{E}}{\bar{P}}$ on those same features (Reaver et al. 2020). In all of these cases, transforming $\frac{\bar{E}}{\bar{P}}$ to $n$ or $w$ adds no new information to the analyses given our finding that the empirical results do not support the conjecture that Eq. (5) and Eq. (6) represent trajectories of individual catchments. For the relationships in Table S1, the parametric Budyko equations essentially

act as (unnecessary) coordinate transformations from Budyko space, with coordinates $\left(\frac{\overline{E_0}}{\bar{P}}, \frac{\bar{E}}{\bar{P}}\right)$, to "Budyko curve space", with coordinates $\left(\frac{\overline{E_0}}{\bar{P}}, n\right)$ or $\left(\frac{\overline{E_0}}{\bar{P}}, w\right)$. Practically, $n$ and $w$ could be eliminated from each of these studies by fitting the proposed models to the estimated values of $\frac{\bar{E}}{\bar{P}}$ directly, bypassing the parametric Budyko framework altogether. The resulting models would likely be easier to interpret, as they would relate catchment biophysical features to a real quantity, either $\frac{\bar{E}}{\bar{P}}$ or $\bar{E}$, rather than to an ambiguous parameter. In short, using the parametric

Budyko equations to estimate $\bar{E}$ from $\bar{P}$ and $\overline{E_0}$ always requires one to first estimate $\bar{E}$; the same is true for estimating *changes* in $\bar{E}$ from *changes* in $\bar{P}$ and $\overline{E_0}$. This severely limits the practical applicability of the parametric Budyko framework.

     In principle, with an appropriate interpretation of the catchment-specific parameter, use of the parametric Budyko framework in landscape hydrology is benign, if unnecessary. However, in practice, even with an

appropriate interpretation of $n$ and $w$, the use of Eq. (5) and (6) in a hydrological analysis will likely have deleterious effects on both the quantitative values of results and their interpretations. The reason for this is that the catchment specific parameter is a poor proxy variable for $\frac{\bar{E}}{\bar{P}}$, since it maps the finite space of $\frac{\bar{E}}{\bar{P}}$ (i.e., 0 to 1) to the infinite spaces of $n$ (i.e., 0 to $\infty$) and $w$ (i.e., 1 to $\infty$). Therefore, as a catchment approaches the water and energy limits in Budyko space, infinitesimal changes in $\frac{\bar{E}}{\bar{P}}$ result in infinitely large changes in the catchment-specific

parameter, allowing for small numerical errors to be dramatically amplified and further confounding relationships based on these transformations (e.g., the relationships in Table S1).



### 4.2.2 Non-uniqueness of the parametric Budyko equations

If the family of curves described by parametric Budyko equations are interpreted as trajectories for catchments undergoing changes in aridity, then each possible parametric Budyko equation contradicts all others, since each give specific but non-equivalent functional forms for the trajectories. Even Eq. (5) and (6), which are generally regarded as essentially interchangeable when using the approximate relationship, $w \approx n + 0.72$ (Yang et al., 2008; Andréassian and Sari, 2019), give non-equivalent trajectories, particularly for small values of the catchment specific parameter. The contradiction between Eq. (5) and (6) alone should cast doubt on the current interpretations of parametric Budyko equations, particularly regarding the physical meaning of explicit curves and the provenance and meaning of the catchment-specific parameter. Moreover, our introduction of Eq. (9) and (10) further illustrates the irreconcilable contradictions between competing parametric Budyko equations.

The parametric Budyko equations described by Eq. (5), (6), (9), and (10) represent four equally valid families of curves (Fig. 4), in that they are all monotonically increasing, concave down, and approach the energy and water limits as $\phi$ approaches 0 and $\infty$, respectively. Curves with constant parameters from each of the four parametric Budyko formulations generally cross and diverge as the aridity index changes (Fig. 5). Traveling along a trajectory with a constant catchment-specific parameter (i.e., $n$, $w$, $q_n$, or $q_w$) in one formulation of the parametric Budyko equations means the parameters of the other three formulations *must* continuously change. Thus, Eq. (5), (6), (9), and (10) directly contradict each other

Of the previously proposed parametric Budyko equations, Eq. (5) and (6) have been the most widely used (e.g., (Donohue et al., 2012; Yang et al., 2007; Yang et al., 2009; Shao et al., 2012; Li et al., 2013; Xu et al., 2013; Cong et al., 2015; Yang et al., 2016; Zhang et al., 2018; Abatzoglou and Ficklin, 2017; Xing et al., 2018a; Zhao et al., 2020; Ning et al., 2020b; Ning et al., 2020a; Li et al., 2020c; Li et al., 2020b; Zhang et al., 2019b; Ning et al., 2019; Bai et al., 2019)). Any of these studies could have justifiably used Eq. (9) or (10) instead, as there is not a clear objective reason to choose any one over the others. However, each equation could lead to substantially different and potentially contradictory results. For example, methods for predicting the sensitivity of rainfall partitioning to changes in aridity index or the catchment-specific parameter (Sect. 2.2) rely on the specified shape of the Budyko curve. The use of Eq. (5) to compute sensitivities would produce substantially different results compared to those produced from Eq. (10). Additionally, methods for attributing changes in rainfall partitioning to anthropogenic and climatic changes (Sect. 2.2) will produce contradictory conclusions when using one parametric Budyko formulation compared to using another.





It is important to note that Eq. (5), (6), (9), and (10) are not the only potential parametric Budyko equations. In fact, the Porporato model (Eq. (8)) can be manipulated into a single-parameter Budyko equation (e.g., Harman et al. (2011); Daly et al. (2019b)). There are likely many more, all equally valid, versions with even starker differences in the shapes of the curves (leading to even larger discrepancies between formulations if the current

5   interpretations of explicit Budyko curves and parametric Budyko equations are maintained). This "equifinality" and non-uniqueness of the parametric Budyko equations is incompatible with the overwhelming current interpretation of the parametric framework and lends support to our contention that the parametric Budyko formulations are better understood as arbitrary coordinate transformations between alternative representations of Budyko space.

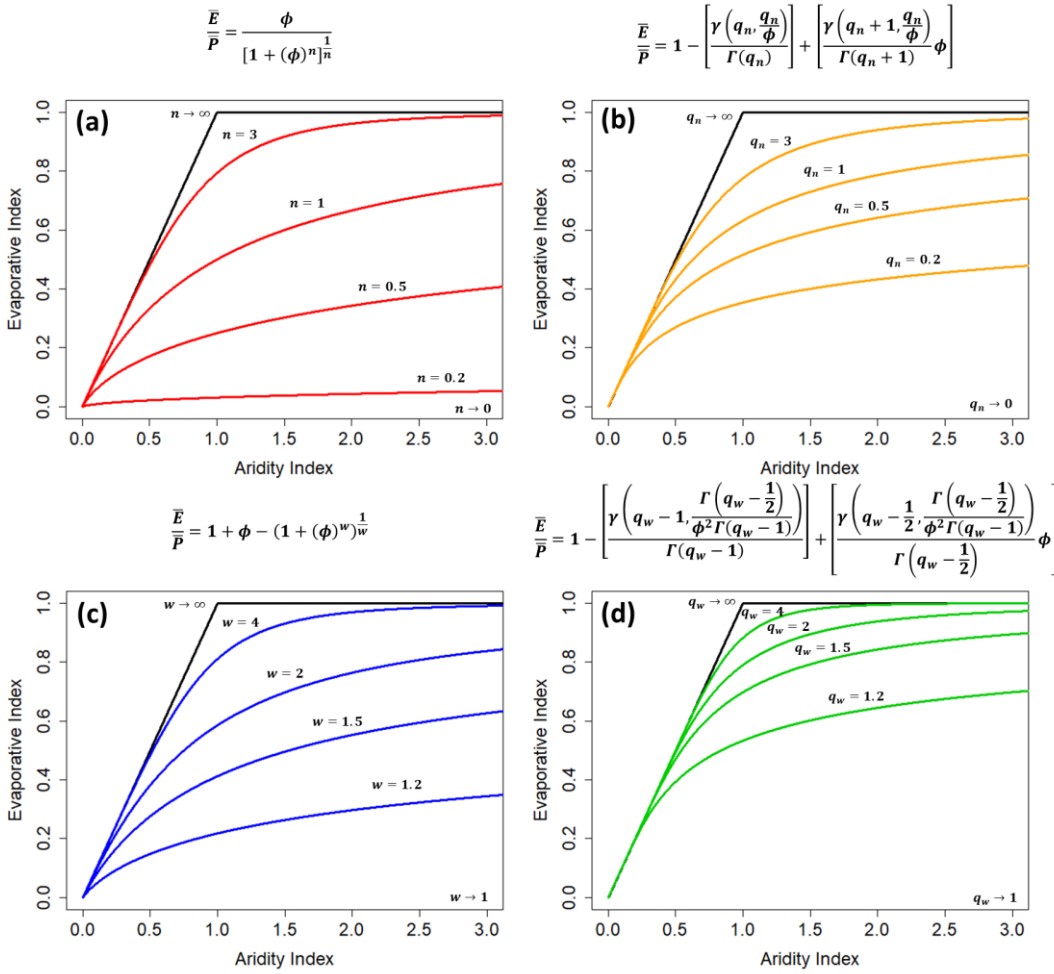

**Figure 4: Illustration of the non-uniqueness of the parametric Budyko equations using (a) Eq. (5), (b) Eq. (9), (c) Eq. (6), and (d) Eq. (10), all of which provide equally valid alternative representations of Budyko space.**



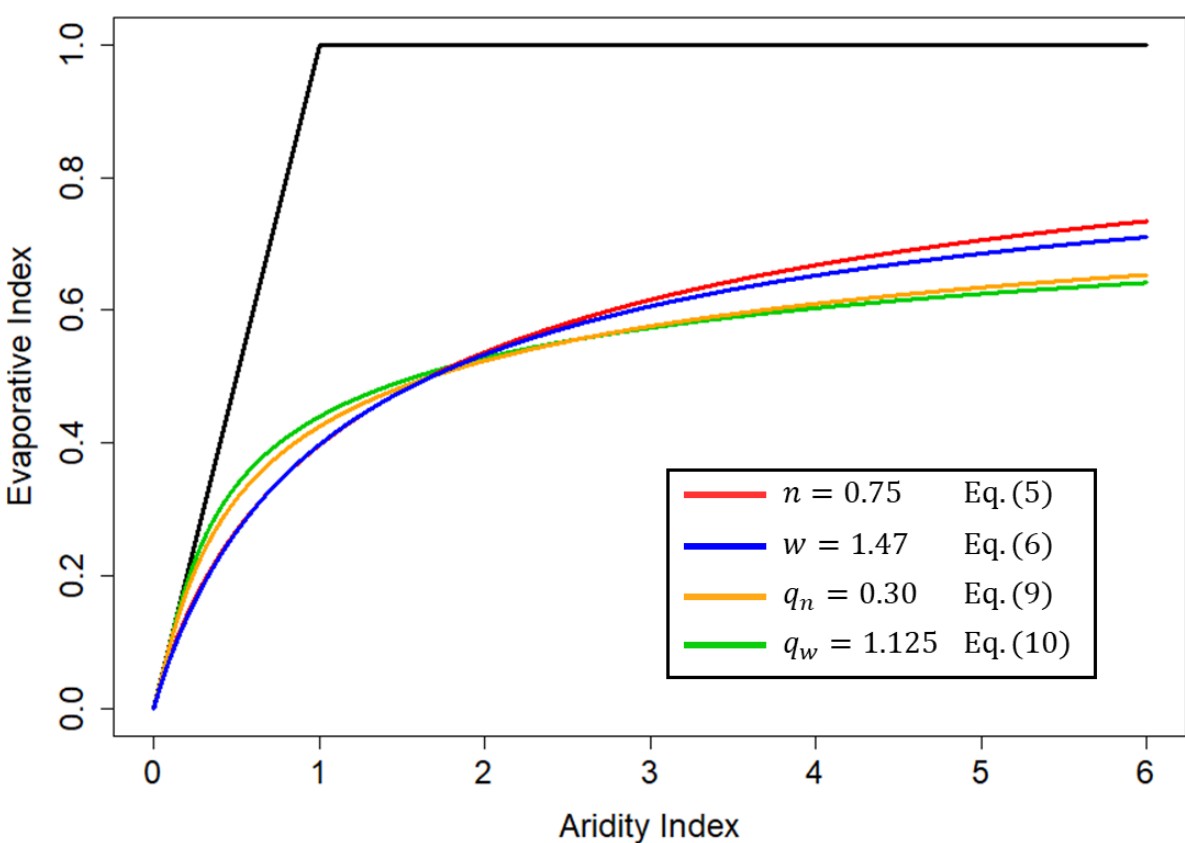

**Figure 5: Illustration of the contradiction between different versions of the four parametric Budyko equations. Constant parameter trajectories, defined by each of the four parametric equations, cross one another. This means that if a catchment has a constant parameter trajectory in one formulation, the parameter must change for the other formulations.**

## 5 Conclusions

The original Budyko hypothesis given in Eq. (2) and the resulting non-parametric curve (e.g., Eq. (3)) provide an overarching framework for understanding catchment hydrology in terms of energy and water balances. As the development of the Budyko framework advanced over the past century, early conceptual tools, such as explicit functional curves, gained considerable influence, resulting in interpretations that are not actually supported by the framework, and which may lead to spurious conclusions. In this study we have revisited, summarized, and





critically evaluated these interpretations, leading to a reinterpretation of explicit Budyko curves and the parametric Budyko equations.

It is apparent from the literature that the prevailing interpretation of explicit Budyko curves ascribes undue physical meaning to the explicit mathematical expression describing the curve. By returning to Budyko's own interpretation of explicit curves, we saw that earlier conceptual frameworks considered the specific choice of functional form to be arbitrary as long as the curves suggested conservation of energy and mass in the humid and arid limits and provided a good representation of the global behavior of multiple catchments across a range of aridities. We reinforce that the general global Budyko curve behavior observed across multiple catchments is a valid, well documented, and physically-driven phenomenon. However, the attribution of physical meaning to the specific functional forms of curves, and explicitly interpreting them as trajectories for catchments undergoing changes in aridity, is an unsupported conjecture. Our tests of this conjecture showed both theoretically and empirically that conceptualizing Budyko curves as trajectories is unjustified. Therefore, as an alternative to using explicit Budyko curves to understand catchment trajectories, we suggest that process-based evapotranspiration models be used. The general Budyko curve behavior can and should be utilized as a global constraint (i.e., validation) for these process-based evapotranspiration models, as any valid model should honor this empirically established behavior when applied to multiple catchments across a range of climates.

A literature review suggests that most current interpretations view the parametric Budyko equations as more general and versatile forms of the non-parametric Budyko equations. We illustrated that the parametric Budyko equations are under-determined, lack predictive power, and are non-unique, merely serving as a coordinate transformation between Budyko space and "Budyko curve space". Coupled to current interpretations of the parametric equations is the idea that the catchment-specific parameter is a lumped quantity that represents the influence of catchment biophysical features on $\frac{\bar{E}}{\bar{P}}$, with many studies in practice treating it as only representing landscape features. We tested the climate independence of the catchment-specific parameter theoretically and demonstrated that its value can change due to climate alone. Given the under-determined nature of the parametric Budyko equations, the catchment-specific parameter is best understood as an arbitrary constant that is effectively a proxy variable for $\frac{\bar{E}}{\bar{P}}$.

The collective results from our analyses suggest that current interpretations of Budyko curve trajectories and the parametric Budyko equations are untenable. We propose that the catchment hydrology community look critically at the well-accepted but unjustified interpretations that are the current standard. This is especially important in view of the recent and growing interest in the application of the Budyko framework.



In closing, we recommend that improved understanding of $\bar{E}$ should emerge from the fundamental physical and biological controls, utilizing the empirically validated global Budyko curve behavior as a constraint, rather than ascribing undue meaning to arbitrary functional forms or ambiguous parameters. As with any empirical relationship, extrapolating the use of the Budyko curve beyond the regime for which is was developed is unjustified without additional evidence. By doing so we risk drawing spurious conclusions about the hydrologic functioning of landscapes. Empirical relationships, such as the Budyko curve, emerge from the underlying physics within a given context, but outside of that context, those relationships are susceptible to losing their physical foundations.

**Supplemental Information**

The supplemental information related to this article is available online at:

**Data availability**

The data used in this manuscript can be obtained from the following locations:
-The CAMELS-GB database (https://doi.org/10.5285/8344e4f3-d2ea-44f5-8afa-86d2987543a9)
-The UKBN2 database (https://nrfa.ceh.ac.uk/benchmark-network) and
(http://nrfa.ceh.ac.uk/sites/default/files/UKBN_Station_List_vUKBN2.0_1.xlsx)
The CAMELS database (https://ral.ucar.edu/solutions/products/camels)

**Author contributions**

NGFR conceived the study, compiled the data, performed the analyses, and drafted the manuscript. All authors contributed in the methodological design, interpretation of results and manuscript preparation.

**Competing interests**

The authors declare no conflicts of interest with respect to the results of this manuscript.

**Acknowledgments**

NGFR acknowledges support from the University of Florida Graduate Fellowship.





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
