# Peer review of "Theoretical and empirical evidence against the Budyko catchment trajectory conjecture"

_Hydrology and Earth System Sciences, 2020_

## Referee Comment (RC1) · Anonymous Referee #1 · 30 Dec 2020

The manuscript entitled "Reinterpreting the Budyko Framework" by Reaver and colleagues highlights several misconceptions regarding recent interpretations of results obtained using the Budyko framework. The authors especially criticize the common assumption that the wealth of functional Budyko curves represent expected trajectories through the Budyko space. By considering a stochastical model and observations from several hundred catchments, it is shown that catchment behavior in time does not follow the predicted trajectories within the Budyko space. The authors further highlight specific parameters used within parametric Budyko equations do not represent catchment-specific biophysical features. The authors thus conclude that Budyko-based results should be interpreted more carefully and thoughtfully.

The manuscript is generally in good shape, overall well structured and well written. The introduction motivates the study and provides an in-depth overview of the recent research within the field. It needs to be acknowledged that this paper addresses a

somewhat heated debate on the interpretation and applicability of the Budyko frame­work in the context of biophysical drivers influencing the terrestrial water and energy balance. However, it is my assessment that the line of arguments and conclusions, as presented in this paper, are mostly adequate. The supporting data and examples seem valid, but I would appreciate a more in-depth justification of several assumptions. I have provided a relatively small number of comments. However, I would also like the authors to consider two more general remarks:

(i) I largely agree with the general conclusions of the paper. However, I know (from my own experience) that the debate on the interpretability of the Budyko framework is somewhat heated. Therefore, I think it needs to be noted that the Budyko framework remains a powerful concept when interpreted and applied correctly. And I don't think that you necessarily "reinterpret" the Budyko framework.

I fully agree that it needs to be acknowledged that there has been a rather large number of recent studies that overinterpreted results. Nonetheless, these studies still present results that are valid and sound within their specific setting. However, any interpretation going beyond these settings is not adequate, which needs to be acknowledged without condemning previous research. You already highlight this in your introduction, but I think you also need to be more careful throughout the rest of the manuscript (see some of the more specific comments below).

In this spirit, I would like to see a more positive evaluation of the Budyko framework per se. I think that the framework, given its adequate application and interpretation, remains super useful. It might thus help to better outline how upcoming Budyko-based research can profit from considering the limitations highlighted in this study. Your con­clusions already provide some suggestions, but I still think that the Budyko framework has more potential besides being a global constraint (p. 25,l. 14), as it can also be applied within well-defined setups.

(ii) Your theoretical example using Porporatos model is neat. However, it is still an

artificial example and also needs to be interpreted as such. You use one model (Porporatos model) to investigate the characteristics of another model (Budyko). Fine, but you need to thoroughly justify that Porporatos model is an appropriate choice in this context: Is the choice of parameter values for the different cases realistic? What kind of conditions do these parameter values represent? Is there any real-world example that would illustrate your choice?

Additionally, as the parameters might be independent within your theoretical modeling framework, they might not be independent under real-world conditions. That also represents another problem of the large number of studies trying to identify biophysical controls. There is no single parameter that controls the partitioning of precipitation into evaporation and runoff. It is rather a convoluted mess of different processes that interact with each other.

Minor comments:

p. 3, l. 7: Please also consider Padron et al., 2017. It provides a comprehensive overview of inconclusive and contradictory evidence obtained from using eq. 6.

p. 4, l. 4-7: It will be helpful to already mention those equations here.

Sec. 2.3: I think it might help to incorporate this section into Sec. 2.1?

p. 10, l. 14: It might be helpful to further explain what you mean by "Budyko-like"?

p. 10, eq. 8: Could you explain in more detail how you estimate the aridity index from this equation?

p. 11, l. 15: Why 2m? Well, this is related to my second major comment above. The choice of these parameter values needs to be justified. What kind of soil characteristics does Z0=2m represent? I know that your overall conclusions won't change when setting Z0=1.9, but it is important to understand what it means and what kind of real-world characteristics your choice represents.

Sec. 3.2: Why don't you include this subsection in the Background part (Sec. 2.)?

p. 14, l. 8-10: Is that true? Are Eqs. 5 and 6 considered the only valid parametric Budyko equations? Do you have more evidence for this statement?

Figure 2: I know it is hard to convey all the necessary information into one Figure, but I have to admit that this one is especially difficult to interpret. The trajectories are a big mess (and to a certain extent this is exactly what you want to highlight here). However, Figure 3a is of more value in this context. If you like to keep Figure 2, maybe consider drawing thinner red lines or introduce some transparency?

p. 25, l. 1-2: Maybe this would be a better title: A reinterpretation of explicit Budyko curves and parametric Budyko equations.

p. 25, l. 20-26: I agree that the interpretation of the parameter representing landscape features is misleading. Calling it a catchment-specific parameter is not justified either. However, even though it is a lumped parameter just existing as a mathematical necessity without any a priori physical interpretation, there might still be an a posteriori physical interpretation. You call the parameter a proxy variable for E/P, which is, in fact, also some sort of physical interpretation. That means, if you assume a constant aridity index and change E/P, the parameter changes as well. Vice versa, if you change the parameter, E/P changes as well. I think the misleading interpretation here is often more related to the assumption that the parameter somehow controls E/P, which is definitely not true.

p. 25, l. 27-28: This statement is too strong in my opinion (see also my first major comment). Any interpretation of obtained results is valid within their specific setting. However, it is the overinterpretation and generalization that is "untenable" (which is a very strong word in this context).

Additional References:

Padrón, Ryan S., Lukas Gudmundsson, Peter Greve, and Sonia I. Seneviratne. 2017.
"Large-Scale Controls of the Surface Water Balance Over Land: Insights From a Systematic Review and Meta-Analysis." Water Resources Research 53 (11): 9659–78. https://doi.org/10.1002/2017WR021215.

---

## Referee Comment (RC2) · Anonymous Referee #2 · 11 Jan 2021

This paper presents interesting tests and perspectives on the Budyko framework. It first argues that there is no theoretical or empirical basis for typical key assumptions in the use of the framework (i.e (i) catchments follow parametric Budyko curves under aridity change, and (ii) the catchment-specific parameter (i.e. $n$ or $w$) is determined by catchment biophysical properties. Subsequently, the paper aims to test these assumptions using outcomes of the Porporato 2004 model, and empirical data.

While the paper addresses two very relevant aspects of the Budyko framework, there seem to be several conceptual limitations that make the results only weakly support the main inferences of the paper, because:

* The approach using the models is that: "We tested the catchment trajectory conjecture by varying the model climatic parameters while holding the landscape parameter constant. If the resulting trajectories are not Budyko curves, the conjecture should be

rejected." This approach assumes that the only relevant climate variable is aridity, but in reality as earlier work has shown (and as the model shows) other climate variables (such as seasonal cycles and P intermittency) also strongly controls water balances. Thus, the observation that the model diverges from the Budyko curves only shows that climate also matters (as is already known) and does not show that catchments do not follow the Budyko trajectory conjecture.

* The approach using the data: It is stated that "this prediction can be tested by comparing actual Budyko space trajectories of reference catchments computed from empirical observations against the expectation from the catchment trajectory conjecture. If the observed reference catchment trajectories are distinct from the expected Budyko curve trajectories, the conjecture should be rejected.". However, there are many other reasons why the trajectories do not follow the Budyko curves. For example, the water balance may not be closed, measurements may be off, climate variables (other than aridity) may also change (since land-cover is the only variable which is controlled for). Therefore it seems somewhat unfair to attribute any anomaly from the curve to solely the Budyko trajectory being wrong, rather than that is also could be caused by any of the other factors.

* To what extent are the methods sensitive to the use of the Hargreaves potential evaporation over any other PET estimate? In theory it seems that the ambiguity of the PET estimate has similar problems as that of catchment specific parameter in the Budyko framework (e.g. suffering from non-uniqueness and potentially crossing trajectories).

Beside these limitations, the paper nicely contrasts the large number of of Budyko studies that "blindly apply Budyko equations", and emphasizes some shortcomings of the framework that are too often ignored. Once the above issues are addressed (and the detailed comments below) I think this paper could make an excellent contribution to the literature.

Detailed comments

L18: "components" or "assumptions"?

L3: "rainfall" should be changed into "precipitation" (as it also includes snow). This change is also recommended at other places where rainfall is stated, but precipitation may be more appropriate.

L30: Note that the Gentine et al. (2012) study excluded most US catchments with loads of snow or out of phase precipitation regimes (i.e. Mediterranean). As a consequence, most scatter was removed, resulting in this interpretation.

L13: It is stated that "we critically reinterpret two key and interrelated components of the current framework:". I am unsure these two things can be called "components". They are rather typical assumptions that people make, but as past authors acknowledge (as referenced by this paper, or as stated above in this review) these assumptions appear largely unfounded, untested, or premature.

L15-17: I appreciate the paper is trying to be gentle towards past research by saying "However, we stress that the aim of this reinterpretation is not to discard the voluminous efforts put forth using current interpretations of the Budyko framework, but rather to recontextualize the conclusions obtained from them". However, your work suggests that all attributions and sensitivity applications will have substantially wrong numbers. This obviously is important "context" but I'd rather say they also cast doubt on many of the past conclusions.

Page 5:

L14: Schreiber, 1904 was not aware yet of the concept of potential evapotranspiration, so I am unsure it is appropriate to cite this work here.
L18: identical ðİŚČİĚ and ðİŘÿ0 ÌĚÌĚÌĚ seems somewhat inaccurate, as it is about the ratio of the two.

Methods

L "(e.g., (" the second layer of brackets seems redundant

Page 11:

Section 3.1.1: This test seems inappropriate for its cause, because climate chararistics other than aridity are varied.

(Also, see main comment above).

Section 3.1.2: Note that similar types of test have been done in https://onlinelibrary.wiley.com/doi/full/10.1002/hyp.9949 and https://www.nature.com/articles/ncomms11603

L18-20: selecting catchments with stable land-use makes the assumption that all other time-varying factors controlling the catchment's water balance (besides aridity) are irrelevant, but this is inaccurate as, for example, seasonal cycles of P can strongly vary between years (and strongly influence the precipitation partitioning).

L20: as a consequence, it is hard to agree with "must be attributed to climatic factors and the catchment trajectory conjecture predicts that their expected trajectories through Budyko space must be Budyko curves". Are the ways to address this critical limitation (given its purpose) to your test?

L28: "ðİŘÿİĚ were calculated from the catchment water balance, ðİŘÿİĚ = ðİŚČİĚ − ðİŚĐİĚ. " is an obvious way to approach the problem, but also known to have issues: https://agupubs.onlinelibrary.wiley.com/doi/10.1029/2020WR027392. What are the potential effects of storage changes (even over 10-y time-scales).

Page 12: "applying moving-average window sizes ranging in annual steps from 1 year to the full length of record." How is it justified to use 1-year windows as these clearly can violate the delta S =~0 assumption?

L25: Why Hargreaves PET, and are there any changes to the results when other PET estimates would be used?

Page 14:

It remains unclear to me what the purpose is of section 3.2.2. (Yes I see WHAT is done, but it seems not really explained WHY this is done).

Results and Discussion

Section 4.1.1.

All these results seem to show that climate variables other than aridity also affect the partitioning of P into Q and E. This seems to be a strange way to test the catchment trajectory conjecture because if the resulting trajectories are not Budyko curves, it just means that climate (other than aridity) also influences the water balances, rather than being a test of the catchment trajectory conjecture. (See main comments).

Section 4.1.2

L6: "their global behaviour". Can this be made more specific (i.e. does it refer only to the long-term mean water balances (e.g. black markers)?).

Questioning that the prevailing interpretations of Budyko curves suggest that the explicit functional forms represent trajectories through Budyko space for individual catchments undergoing changes in aridity index has also been discussed in https://onlinelibrary.wiley.com/doi/10.1002/hyp.13958 and tested in https://www.nature.com/articles/ncomms11603. This may be worth acknowledging.

Figure 2: is there any way to better visualise what is going on here? One minor change (that will not resolve all issues) may be to make the x-axis on a log-scale. This avoids

[Figure]

that humid catchments are all condensed in a tiny part of the left side of the figure (whereas arid catchments are spread out at the right-hand side).

L4.2.2.

please specify that it is the common interpretation not ALL interpretations in "should cast doubt on the current interpretations of parametric Budyko equations,"

Conclusions

"We suggest that process-based evapotranspiration models be used" Note that this is consistent with earlier works: e.g. https://doi.org/10.1029/94WR00586, https://agupubs.onlinelibrary.wiley.com/doi/full/10.1029/2005WR004606, etc).

"The general Budyko curve behavior can and should be utilized as a global constraint". The "should" seems a bit odd as there will be many instances in which there will be better/more data available than the Budyko curve to constrain models.

---

## Short Comment (SC1) · 29 Jan 2021

I have eagerly read this manuscript and am welcoming of a new perspective on the Budyko curve. While the theoretical understanding of the curve has been continuing to grow, my perspective is that actual improvements to the performance of the model in hydrological applications stalled a few decades ago (my own work included!). It would be great to kick-start that process again. So I am glad to read a critical appraisal of the use of the Budyko curves.

I have one comment to make on this manuscript, which relates to the claim that catchments in reality don't follow Budyko-like curves. The authors show this using catchment data from the US and the UK (Figure 2). The data show that the catchments do generally follow a curve when using the long-term averages but not when using time-series (looking at a catchment through time). This is an expected result due to the underlying

assumption in the framework that precipitation is the \*only\* supply of water. This is often interpreted in terms of catchments needing to be in steady state.

Given this inherent model assumption, one should expect time-series catchment data NOT to follow a single curve. Hence, when the authors say that

"...we can conclude that individual catchments do not generally or consistently follow Budyko curve trajectories as posited by the catchment trajectory conjecture..." (page 17 line 8)

what the authors have (re)discovered is what happens when one violates a key model assumption.

Hence, their assertion that

"...this conjecture in hydrological analyses (e.g., precipitation partitioning sensitivity and causal attribution to anthropogenic and climatic impacts) will likely introduce significant errors and may lead to spurious conclusions."

seems to be difficult to support from their empirical test but also seems unfair both to the model itself and to those in the community who apply the model in accordance to its inherent limitations.

Is it not the case that only time-series (daily, weekly, yearly etc) hydrological data that have water storage change effects accounted for can be used to test, empirically, whether the catchment specific parameter is temporally constant?

Might it be that water storage provides the link between the analytical meaning of the parameter and the physical understanding of how individual catchments have different E for the same P and Ep? That is, when a stored water term is included as a water supply term alongside P, then it introduces a catchment-specific and time-dependent term into the model fundamentals and into your analytical analysis?

584, 2020.

---

## Author Comment (AC1) · 6 Apr 2021

We thank the reviewer for these helpful comments. Reviewer comments are listed below, along with our response to each. In some cases, we describe the proposed revisions to the manuscript (with line numbers) or refer to proposed revisions described in our responses to the other reviewers, but we recognize that the revised manuscript is requested in a subsequent step.

**Comment 1:**

**The manuscript entitled "Reinterpreting the Budyko Framework" by Reaver and colleagues highlights several misconceptions regarding recent interpretations of results obtained using the Budyko framework. The authors especially criticize the common assumption that the wealth of functional Budyko curves represent expected trajectories through the Budyko space. By considering a stochastical model and observations from several hundred catchments, it is shown that catchment behavior in time does not follow the predicted trajectories within the Budyko space. The authors further highlight specific parameters used within parametric Budyko equations do not represent catchment-specific biophysical features. The authors thus conclude that Budyko-based results should be interpreted more carefully and thoughtfully.**

Response 1:

We thank the reviewer for an accurate representation, though we note that another outcome of this work is to illustrate that the non-uniqueness of the parametric Budyko equations (i.e., there are several equally valid single parameter equations with different functional forms) fundamentally contradicts many recent interpretations of the parametric Budyko framework.

**Comment 2:**

**The manuscript is generally in good shape, overall well structured and well written. The introduction motivates the study and provides an in-depth overview of the recent research within the field. It needs to be acknowledged that this paper addresses a somewhat heated debate on the interpretation and applicability of the Budyko framework in the context of biophysical drivers influencing the terrestrial water and energy balance. However, it is my assessment that the line of arguments and conclusions, as presented in this paper, are mostly adequate. The supporting data and examples seem valid, but I would appreciate a more in-depth justification of several assumptions.**

Response 2:

We thank the reviewer for the kind words and the positive assessment of the manuscript. We address the request for additional justification of assumptions in subsequent responses.

**Comment 3:**

**I have provided a relatively small number of comments. However, I would also like the authors to consider two more general remarks:**

**(i) I largely agree with the general conclusions of the paper. However, I know (from my own experience) that the debate on the interpretability of the Budyko framework is somewhat heated. Therefore, I think it needs to be noted that the Budyko framework remains a powerful concept when interpreted and applied correctly. And I don't think that you necessarily "reinterpret" the Budyko framework. I fully agree that it needs to be acknowledged that there has been a rather large number**

**of recent studies that overinterpreted results. Nonetheless, these studies still present results that are valid and sound within their specific setting. However, any interpretation going beyond these settings is not adequate, which needs to be acknowledged without condemning previous research. You already highlight this in your introduction, but I think you also need to be more careful throughout the rest of the manuscript (see some of the more specific comments below).**

**In this spirit, I would like to see a more positive evaluation of the Budyko framework per se. I think that the framework, given its adequate application and interpretation, remains super useful. It might thus help to better outline how upcoming Budyko-based research can profit from considering the limitations highlighted in this study. Your conclusions already provide some suggestions, but I still think that the Budyko framework has more potential besides being a global constraint (p. 25,l. 14), as it can also be applied within well-defined setups.**

Response 3:

We agree that the Budyko framework is a powerful concept and have expressed its validity in the manuscript (page 25 lines 8-9, page 25 lines 14-16). However, we take the reviewer's point and therefore propose specific edits (listed at the end of the response) to strengthen this concept in the manuscript as well as outline useful directions for future Budyko-based research (see also proposed edits in the response to **Reviewer 2 Comment 25**). In addition, we do not intend to condemn previous research and explicitly attempt to convey this sentiment in the manuscript (page 3 lines 15-17) (see also our responses to **Reviewer 2 Comment 10** and **SC1 Comment 2**). However, we acknowledge the need to be more specific and state that we believe that only the "original" Budyko framework (i.e., the observation that the aggregate behavior of multiple catchments consistently produce a distinctive pattern in Budyko space) remains intrinsically useful. Specifically, the emergent aggregate Budyko curve pattern provides: (1) an empirical constraint for process-based model validation and theoretical investigations and into catchment hydraulic processing; (2) a practical constraint for process-based model calibration; and (3) allows for probabilistic predictions of $\bar{E}$ and $\bar{Q}$ as well as changes in $\bar{E}$ and $\bar{Q}$ for ungauged basins with limited data. Used in these contexts, the Budyko framework is powerful and useful.

However, based on our review of the literature (and the analyses and arguments presented in this manuscript), we do not agree that the parametric Budyko framework is intrinsically useful to the catchment hydrology community. Since the functional forms of the parametric Budyko equations are not empirically valid (see Sections 3.1.2 and 3.2.2 in the main manuscript), these equations are literally only an arbitrary mathematical transform of the evaporative index. As we state in the manuscript (page 21 lines 18-19), with this mathematically sound interpretation, use of the parametric framework in hydrological applications is theoretically benign, but it is unnecessary and can easily amplify errors. Essentially all studies that use single-parameter Budyko equations could have conducted the same analyses on $\bar{E}$ or the evaporative index instead.

A good example of a study that used the parametric Budyko framework robustly, appropriately, and in a manner which did not impact its outcome is Greve et al. (2020). This study calibrated several parameters of a global hydrological model using a parametric Budyko framework-based constraint. This methodology significantly improved the global hydrological model's performance compared to its uncalibrated version. However, as we argue in our manuscript (see page 21 lines 7-17), Greve et al.

(2020) could have compared the simulated and empirical joint distributions of $\frac{\bar{E}}{P}$ and $\overline{\frac{E_0}{P}}$ directly, without using the distributions of the catchment-specific parameter (ω) as an intermediary. The parametric Budyko framework could be easily removed from this analysis without changing the results. Thus, while the use of parametric Budyko framework did not change the outcome of the study, it acted as an unnecessary mathematical transform. Similar benign situations occur for essentially all appropriate uses of the parametric Budyko framework, while inappropriate uses often lead to spurious results.

Despite this critique, we also stress that, even for prior research that utilized inappropriate interpretations of the parametric framework, both the intent and much of the effort of previous research can be preserved. For example, as we state in the manuscript (page 21 lines 9-14), any study that has related $n$ or $w$ to catchment biophysical features could remove the parametric framework from their analyses and use their same analytical tools to relate $\bar{E}$ or $\frac{\bar{E}}{P}$ to biophysical features directly. This would preserve most of the analyses of such studies (i.e., same analytical methods) as well as the intent (i.e., understanding the interactions between $\bar{E}$ and catchment biophysical features).

To strengthen our representation of the validity of the "original" Budyko framework in the manuscript, highlight how the intent and effort of previous research can be preserved, and to better outline useful directions for future Budyko-based research, we propose the following edits:

1) Add the following sentence to page 3 line 17:
   *"Additionally, we emphasize that the Budyko framework based on the curve-like clustering pattern observed across multiple catchments is a powerful and useful concept when used appropriately and with proper context."*

2) Edit the sentence page 25 lines 14-16 to the following:
   *"Additionally, to be a valid representation of catchment evapotranspiration, process-based models need to able to reproduce the empirically established general Budyko curve behavior (i.e., nonparametric) when applied to multiple catchments across a range of climates. As such, the general Budyko curve behavior can serve as a global constraint (i.e., calibration or validation) in the application of such models, e.g., Greve et al. (2020). Furthermore, while the parametric Budyko framework lacks predictive power, the nonparametric framework allows for probabilistic predictions of $\bar{E}$ and $\bar{Q}$ as well as changes in $\bar{E}$ and $\bar{Q}$ for ungauged basins. Within these contexts, the nonparametric Budyko framework is a useful conceptualization. "*

3) Add the following to the end of Section 4.2.1:
   *"While the acknowledgment of the proxy nature of the catchment-specific parameter and $\frac{\bar{E}}{P}$ casts doubt on the specific conclusions of previous parametric Budyko-based research, we note that both the intent and much of the effort of many such studies can be preserved. For example, studies that related $n$ or $w$ to catchment biophysical features using various analytical tools could employ the same methods relate $\bar{E}$ or $\frac{\bar{E}}{P}$ to biophysical features directly. Doing so would preserve most of the analyses of such studies (i.e., near identical methods) as well as their intent (i.e., understanding the relationship between $\bar{E}$ and catchment biophysical features)."*

**Comment 4:**

**(ii) Your theoretical example using Porporatos model is neat. However, it is still an artificial example and also needs to be interpreted as such. You use one model (Porporatos model) to investigate the characteristics of another model (Budyko). Fine, but you need to thoroughly justify that Porporatos model is an appropriate choice in this context: Is the choice of parameter values for the different cases realistic? What kind of conditions do these parameter values represent? Is there any real-world example that would illustrate your choice?**

**Additionally, as the parameters might be independent within your theoretical modeling framework, they might not be independent under real-world conditions. That also represents another problem of the large number of studies trying to identify biophysical controls. There is no single parameter that controls the partitioning of precipitation into evaporation and runoff. It is rather a convoluted mess of different processes that interact with each other.**

Response 4:

We agree that the Porporato model is an artificial example (i.e., model investigating a model) and has some limitations (e.g., assumes $E_0$ is constant), though we do not believe that these properties influence the general conclusions from the theoretical test. Our primary justifications for using the Porporato model were its simplicity (see page 10 lines 20-25 and page 11 line 1), physically-based nature (see page 10 lines 7-9), and its previous use within the Budyko framework literature (see page 10 lines 5-7). However, we take the reviewer's point that the appropriateness of the Porporato model's use could be further justified in the manuscript and propose to do so through the suggested edits in this response.

We understand the reviewer's concerns regarding the model's parameter values and whether they are reflective of conditions in real catchments, as models should typically reflect reality. However, for the Porporato model, this is somewhat unimportant since all parameters appear in ratios. This means the individual parameter value magnitudes are less important than the relative magnitudes between parameters. This flexibility allowed us to choose values that were more useful for illustrative purposes as opposed to precise values reflective of a particular catchment conditions. Specifically, the values of the parameters were primarily chosen for the following reasons: (1) to maintain the simplicity of the illustrative examples (e.g., using integers); (2) to allow all of the test cases to be expressed through a single functional relationship (see page 11 lines 12-15); (3) to produce trajectories that would be visually informative (e.g., not restricted to a small portion of Budyko space such as being compressed at the water and energy limits). However, we note and emphasize that choosing values representing particular catchment conditions (e.g., an effective rooting depth of 0.374 m) would not change the results (since the same trajectories could be produced by adjusting other parameters), but the simplicity and clarity of the test cases would be lost. To reflect this view in the manuscript better, we propose to add the following paragraph to the end of Section 3.1.1:

*"The effective climate and landscape parameters exclusively appear in ratios within the Porporato model. This means only the relative magnitude between parameters is important. Therefore, for our test cases, we chose parameter values to maintain illustrative simplicity, allow all test case to be expressed in a unified functional form, and to produce visually informative trajectories not restricted to a small portion of Budyko space. This choice does not impact the conclusions of the test cases, since the ratio*

*nature of the model's parameters means the exact same trajectories can be made from infinitely many different parameter combinations."*

The reviewer rightly points out that there may be possible dependencies between model parameters and that the test cases should correspond to real-world examples. We note that for the variable parameter test cases, we explicitly considered the possible dependencies of the climate parameters (i.e., the relationships between $\eta$ and $\psi$) (see page 11 lines 12-15) and chose test cases that are reflective of real-world conditions (see page 11 lines 7-12 and the references included, Trenberth (2011) and Fischer et al. (2014)).

**Comment 5:**

**Minor comments:**

**p. 3, l. 7: Please also consider Padron et al., 2017. It provides a comprehensive overview of inconclusive and contradictory evidence obtained from using eq. 6.**

Response 5:

We agree that Padrón et al. (2017) should be included and propose to add the following sentences starting at page 3 line 5:

*"Furthermore, Padrón et al. (2017) undertook a comprehensive overview of the wide variety of biophysical features proposed to control the catchment-specific parameter, finding that most proposed features did not actually correlate with the parameter and the types of features that were correlated varied sustainably between climatic regions."*

**Comment 6:**

**p. 4, l. 4-7: It will be helpful to already mention those equations here.**

Response 6:

We agree that it would be helpful to have already introduced both of the parametric equations for readers who are uninitiated to the catchment hydrology literature. However, doing so would significantly alter the flow and structure of the introduction and background, and such readers are only a portion of the intended audience. Given the apparent contentious status of the topics covered into this manuscript (as the reviewer notes in **Comment** 2 and **Comment 3**), we have intentionally structured our introduction to ensure that the reader is informed of the full context, implications, and summary of the manuscript's contents before being introduced to the more detailed background, analyses, and discussions. We propose to keep this structure, however we understand the reviewer's point and therefore propose to edit the sentence on page 4 lines 4-5 to:

*"We argue and demonstrate herein that the two widely accepted parametric Budyko equations (i.e., those derived in Zhang et al. (2004) and Yang et al. (2008)) are non-unique, meaning they are only two of many possible single-parameter Budyko equations."*

**Comment 7:**

**Sec. 2.3: I think it might help to incorporate this section into Sec. 2.1?**

Response 7:

We understand that "Budyko's interpretation of explicit curves" could naturally be classified as part of an "Overview of the Budyko hypothesis and equations", however, we specifically placed it after "Current interpretations of explicit Budyko curves and the parametric framework" to contrast how current interpretations have evolved (or strayed) from the "original" intent of explicit curves. This section provides the start of our reinterpretation (or "retrospective interpretation") from the current state of the framework. Therefore, we believe it is important to highlight Budko's interpretation independently and following the introduction of current interpretations.

**Comment 8:**

**p. 10, l. 14: It might be helpful to further explain what you mean by "Budyko-like"?**

Response 8:

We propose changing p. 10, l. 13 to:

*"We first write the model of Porporato et al. (2004) in a form which can be plotted in Budyko space,…"*

**Comment 9:**

**p. 10, eq. 8: Could you explain in more detail how you estimate the aridity index from this equation?**

Response 9:

We propose adding the following sentence at Line 1 on Page 11 to make this clearer:

*"The ratio of $\psi$ and $\eta$ is the aridity index, $\phi = \frac{\overline{E_0}}{\overline{P}} = \frac{\overline{E_0}}{\alpha\lambda} = \frac{\psi}{\eta}$."*

**Comment 10:**

**p. 11, l. 15: Why 2m? Well, this is related to my second major comment above. The choice of these parameter values needs to be justified. What kind of soil characteristics does Z0=2m represent? I know that your overall conclusions won't change when setting Z0=1.9, but it is important to understand what it means and what kind of real-world characteristics your choice represents.**

Response 10:

We have addressed this comment in our response to **Comment 4.**

**Comment 11:**

**Sec. 3.2: Why don't you include this subsection in the Background part (Sec. 2.)?**

Response 11:

The Background section currently only contains existing information in the literature. We believe the content of Section 3.2 is a new contribution and therefore the content belongs in the methodological and discussion portions of the manuscript.

**Comment 12:**

**p. 14, l. 8-10: Is that true? Are Eqs. 5 and 6 considered the only valid parametric Budyko equations? Do you have more evidence for this statement?**

Response 12:

We thank the reviewer for addressing this point. Upon review of our original claim, which appears to be a common theme in the literature, we now agree that there are other single parameter functional forms that also satisfy the uniqueness requirement, e.g., a form of the equation introduced in Porporato et al. (2004). As such, we have proposed edits to Section 3.2.2 in our response to **Reviewer 2 Comment 20** to focus more on the properties that are typically used to justify the validity of Eq. (5) and (6) and highlight how these properties are not unique to them.

**Comment 13:**

**Figure 2: I know it is hard to convey all the necessary information into one Figure, but I have to admit that this one is especially difficult to interpret. The trajectories are a big mess (and to a certain extent this is exactly what you want to highlight here). However, Figure 3a is of more value in this context. If you like to keep Figure 2, maybe consider drawing thinner red lines or introduce some transparency?**

Response 13:

We agree with the suggestions of the reviewer and those provided in **Reviewer 2 Comment 23**. We propose to reduce the thickness of the red lines and logarithmically (base 10) scale the abscissa and modify the text referencing this figure accordingly. The resulting figure is:

[Figure]

**Figure 2: Semi-log plot of the Budyko space locations (black dots) of the 728 UK and US reference catchments and their corresponding expected Budyko curve trajectories, Eq. (5) (gray dashed curves) and 10-year average actual trajectory realizations (red solid curves). The global behavior of the catchments and their actual trajectories generally agrees with the the non-parametric Budyko, Eq. (3) (blue solid curve) but not the expected parametric Budyko curve trajectories.**

**Comment 14:**

**p. 25, l. 1-2: Maybe this would be a better title: A reinterpretation of explicit Budyko curves and parametric Budyko equations.**

Response 14:

We agree this title provides a more focused representation of the contents of the manuscript. Therefore, we propose adopting a modified version of it in the revised manuscript:

*"A reinterpretation of explicit curves and parameters of the Budyko framework"*

**Comment 15:**

**p. 25, l. 20-26: I agree that the interpretation of the parameter representing landscape features is misleading. Calling it a catchment-specific parameter is not justified either. However, even though it is a lumped parameter just existing as a mathematical necessity without any a priori physical interpretation, there might still be an a posteriori physical interpretation. You call the parameter a proxy variable for E/P, which is, in fact, also some sort of physical interpretation. That means, if you assume a constant aridity index and change E/P, the parameter changes as well. Vice versa, if you change the parameter, E/P changes as well. I think the misleading interpretation here is often more related to the assumption that the parameter somehow controls E/P, which is definitely not true.**

Response 15:

As we state in the manuscript (page 13 lines 15-29), in order for the "catchment-specific parameter" to have *a posteriori* physical interpretation, the associated functional form (i.e., Eqs. (5) or (6)) would have to be empirically valid. But, because our empirical test of the catchment trajectory conjecture refutes the parametric Budyko curves' empirical validity, making Eqs. (5) and (6) under-determined (i.e., 1 equation with 2 unknowns, $\frac{\bar{E}}{P}$ and $n$ or $w$), we conclude (page 20 lines 16-29) that the "catchment-specific parameter" does not have an interpretation independent of $\frac{\overline{E_0}}{P}$ and $\frac{\bar{E}}{P}$.

We say the "catchment-specific parameter" is a proxy for $\frac{\bar{E}}{P}$ since, in practice, the values of $\frac{\overline{E_0}}{P}$ (usually taken as a known quantity) and $\frac{\bar{E}}{P}$ ***always*** determine the "catchment-specific parameter". For real catchments, the association between $\frac{\bar{E}}{P}$ and the "catchment-specific parameter" is one-way; $\frac{\overline{E_0}}{P}$ and $\frac{\bar{E}}{P}$ are used to compute $n$ or $w$. It is never the case that $\frac{\overline{E_0}}{P}$ and $n$ or $w$ are used to compute $\frac{\bar{E}}{P}$. Therefore, we agree with the reviewer that the "catchment-specific parameter" does not control $\frac{\bar{E}}{P}$, however $\frac{\bar{E}}{P}$ completely controls the value of the "catchment-specific parameter" (page 21 lines 1-7).

**Comment 16:**

**p. 25, l. 27-28: This statement is too strong in my opinion (see also my first major comment). Any interpretation of obtained results is valid within their specific setting. However, it is the overinterpretation and generalization that is "untenable" (which is a very strong word in this context).**

Response 16:

We have largely addressed the philosophical underpinnings of this comment in our response to **Comment 3**. While we believe that the current sentence is accurate, we take the reviewer's point about the strong wording of this statement. Therefore, we propose to edit the sentences on page 25 lines 27-29 to:

*"The collective results from our analyses suggest that current commonly held interpretations of Budyko curve trajectories and the parametric Budyko equations are unsupported. We propose that the catchment hydrology community look critically at the well-accepted but unjustified interpretations that are the current commonly held standard."*

**References:**

Fischer, E. M., Sedláček, J., Hawkins, E., and Knutti, R.: Models agree on forced response pattern of precipitation and temperature extremes, Geophysical Research Letters, 41, 8554-8562, 10.1002/2014gl062018, 2014.
Greve, P., Burek, P., and Wada, Y.: Using the Budyko Framework for Calibrating a Global Hydrological Model, Water Resources Research, 56, e2019WR026280, 10.1029/2019wr026280, 2020.
Padrón, R. S., Gudmundsson, L., Greve, P., and Seneviratne, S. I.: Large-Scale Controls of the Surface Water Balance Over Land: Insights From a Systematic Review and Meta-Analysis, Water Resources Research, 53, 9659-9678, 10.1002/2017wr021215, 2017.
Porporato, A., Daly, E., and Rodriguez-Iturbe, I.: Soil water balance and ecosystem response to climate change, The American Naturalist, 164, 625-632, 2004.
Trenberth, K. E.: Changes in precipitation with climate change, Climate Research, 47, 123-138, 10.3354/cr00953, 2011.
Yang, H., Yang, D., Lei, Z., and Sun, F.: New analytical derivation of the mean annual water-energy balance equation, Water Resources Research, 44, n/a-n/a, 10.1029/2007wr006135, 2008.
Zhang, L., Hickel, K., Dawes, W. R., Chiew, F. H. S., Western, A. W., and Briggs, P. R.: A rational function approach for estimating mean annual evapotranspiration, Water Resources Research, 40, n/a-n/a, 10.1029/2003wr002710, 2004.

---

## Author Comment (AC2) · 6 Apr 2021

Please see the PDF supplement for the response to Anonymous Referee #2

Please also note the supplement to this comment: https://hess.copernicus.org/preprints/hess-2020-584/hess-2020-584-AC2-supplement.pdf

---

## Author Comment (AC3) · 6 Apr 2021

We thank the reviewer (Dr. Randall Donohue) for these helpful comments. Reviewer comments are listed below, along with our response to each. In some cases, we describe the proposed revisions to the manuscript (with line numbers) or refer to proposed revisions described in our responses to the other reviewers, but we recognize that the revised manuscript is requested in a subsequent step.

**Comment 1:**

**I have eagerly read this manuscript and am welcoming of a new perspective on the Budyko curve. While the theoretical understanding of the curve has been continuing to grow, my perspective is that actual improvements to the performance of the model in hydrological applications stalled a few decades ago (my own work included!). It would be great to kick-start that process again. So I am glad to read a critical appraisal of the use of the Budyko curves.**

Response 1:

We thank the reviewer for his interest in our manuscript and appreciate the kind words.

**Comment 2:**

**I have one comment to make on this manuscript, which relates to the claim that catchments in reality don't follow Budyko-like curves. The authors show this using catchment data from the US and the UK (Figure 2). The data show that the catchments do generally follow a curve when using the long-term averages but not when using time-series (looking at a catchment through time). This is an expected result due to the underlying assumption in the framework that precipitation is the \*only\* supply of water. This is often interpreted in terms of catchments needing to be in steady state.**

**Given this inherent model assumption, one should expect time-series catchment data NOT to follow a single curve. Hence, when the authors say that**

**"...we can conclude that individual catchments do not generally or consistently follow Budyko curve trajectories as posited by the catchment trajectory conjecture..." (page 17 line 8)**

**what the authors have (re)discovered is what happens when one violates a key model assumption.**

**Hence, their assertion that**

**"...this conjecture in hydrological analyses (e.g., precipitation partitioning sensitivity and causal attribution to anthropogenic and climatic impacts) will likely introduce significant errors and may lead to spurious conclusions."**

**seems to be difficult to support from their empirical test but also seems unfair both to the model itself and to those in the community who apply the model in accordance to its inherent limitations.**

Response 2:

First, we wish to make it clear that our intention is not to be unfair to the Budyko framework, nor to those who apply it. Our conclusions about the fundamental limitations of the parametric Budyko equations emerged from a genuine interest in the framework and a careful study of the catchment hydrology literature while attempting to improve the biophysical understanding of the catchment-specific parameters. After realizing the non-transferability and under-determined nature of the parametric Budyko equations in our own applications, we decided it would be beneficial to bring these

issues to the larger catchment hydrology community, particularly given the large number of recent papers using the parametric framework. We have attempted to convey a message of recontextualization of prior results in the manuscript (e.g., page 3 lines 15-17, page 21 lines 11-14). However, we can likely improve our representation of this message, as also suggested by **Reviewer 1.** We will address this by explicitly illustrating how the intent and efforts of prior work can be maintained if an appropriate interpretation of the parametric Budyko framework is applied (see proposed edits to the manuscript in our responses to **Reviewer 1 Comment 3** and **Reviewer 1 Comment 16**).

Next, we thank the reviewer for his accurate representation of the results from our empirical test of the catchment trajectory conjecture. Specifically, we found that the long-term behavior of hundreds of US and UK reference catchments generally follow the non-parametric Budyko curve when taken as an ensemble, however the behavior of individual catchments over time do not follow the conjectured parametric Budyko curves. The reviewer suggests, however, that time-series catchment data should not necessarily follow a single explicit curve, since such data violate the underlying steady state assumption of the Budyko approach. He therefore suggests that our conclusions about the catchment trajectory conjecture are not supported by our empirical test.

However, the reviewer's interpretation neglects two central elements of empirical test methodology. Specifically, we used only reference catchments (i.e., stable catchment characteristics) and accounted for potential non-steady state storage impacts by testing catchments' actual trajectories through Budyko space for essentially all possible temporal averaging windows (see further details in our response to **Comment 3**). We thus believe that we fairly tested the important assumptions and interpretations of the parametric Budyko framework—and found them to be unsupported for individual catchments. While the ensemble behavior of many catchments generally follows the non-parametric Budyko curves, our evidence suggests the trajectories of individual catchments are not specific parametric curves.

To be clear, we agree that not accounting for storage dynamics is a violation of the Budyko framework's underlying assumption. However, we note that the catchment trajectory conjecture and methods derived from it specifically suggest that the temporal evolution of a catchment (i.e., its time series) will follow a particular Budyko curve under changes in aridity index. For example, the two derived methods that the reviewer quotes from our manuscript, precipitation partitioning sensitivity and causal attribution to anthropogenic and climatic impacts, are fundamentally based on the idea of change in a catchment's Budyko space trajectory over time. Combined with the reviewer's **Comments 3 and 4** below, we thus conclude that the reviewer is suggesting that time series data will follow a particular Budyko curve, as long as the catchment properties remain unchanged and the storage dynamics are properly accounted for (or the assumption of steady state is not violated). We agree with this interpretation of required conditions (i.e., stable catchment characteristics and steady state storage) and contend that our empirical test methodology meets these requirements and therefore provides a robust assessment of the claims of the catchment trajectory conjecture.

**Comment 3:**

**Is it not the case that only time-series (daily, weekly, yearly etc) hydrological data that have water storage change effects accounted for can be used to test, empirically, whether the catchment specific parameter is temporally constant?**

Response 3:

Yes, in order to quantify $\bar{E}$ accurately for a catchment (which is required to test if the catchment-specific parameter is temporally constant), the remaining components of the water balance must be known since, $\bar{E} = \bar{P} - \bar{Q} - \overline{\Delta S}$. While the 728 references catchments used in our empirical analysis had daily time series of $P$ and $Q$ (which can be averaged over the time interval of interest to obtain $\bar{P}$ and $\bar{Q}$), they lacked estimates of $\Delta S$, as is the case for the vast majority of available catchment hydrology data. We recognized this potential limitation in our analysis and addressed it explicitly by testing the catchment trajectory conjecture for all possible relevant realizations of each catchment's actual trajectory through Budyko space (page 11 lines 26-29 and page 12 lines 1-3). Specifically, we computed time-varying $\bar{P}$, $\overline{E_0}$, and $\bar{E}$ by applying moving-average window sizes ranging in annual steps from 1 year to the full length of record (which ranged between 12 and 45 years for all catchments).

It should be expected that above some threshold averaging window size (e.g., 10-, 20-, 30-year average window), changes in catchment storage average to zero ($\overline{\Delta S} \sim 0$). This has been shown to be the case for catchments across Earth (Han et al., 2020), with 71% of catchments reaching steady state with a 10-year averaging window and 94% reaching steady state with a 30-year averaging window. However, even if this threshold behavior does not apply universally, it should be expected that, for some averaging windows and some catchments, steady state conditions would be present (i.e., $\overline{\Delta S} \sim 0$). In either case (threshold behavior or not), testing all the relevant averaging windows for all catchments allows for a robust test of the catchment trajectory conjecture. In the case of threshold steady state behavior, if the catchment trajectory conjecture is correct, actual and expected Budyko space trajectories for a catchment would be consistently and statistically indistinguishable once the averaging window reaches a sufficient length (see page lines and page 12 lines 4-13). Statistically, this means that the frequency at which actual and expected Budyko space trajectories were found to be statistically indistinguishable would be higher than expected at random (i.e., more than 5% of all actual vs. expected trajectories would be statistically indistinguishable at a significance level of 0.05). An elevated frequency of statistical similarity would also occur if catchments were only rarely in steady state. The reason for this elevation is that the number of statistically similar trajectories from averaging windows where $\overline{\Delta S} \sim 0$ would be in addition to the number of statistically similar trajectories expected from random chance.

Critically, the results of our empirical test presented in the main text show that the catchment trajectory conjecture is not supported. Out of the 24,501 actual trajectory realizations, 23,231 (95%) were found to have consistent differences (p-value < 0.05) from their "expected" trajectories, while only 1270 (5%) were found to be statistically indistinguishable. This proportionality is exactly what would be expected due to random chance. We emphasize that our methodology directly addresses the (very common) lack of knowledge of catchment storage dynamics that the reviewer points out, and it thus provides a robust test of the catchment trajectory conjecture. However, we believe we can be clearer on this point in the manuscript, and so we suggest the following edits:

1) Change the section starting on page 11 line 26 and ending on page 12 line 7 to,

*"For a given reference catchment, estimates of $\bar{P}$ and $\overline{E_0}$ were obtained from daily records of $P$ and $E_0$, while estimates of $\bar{E}$ were calculated from the catchment water balance, $\bar{E} = \bar{P} - \bar{Q}$, which assumes impacts from storage dynamics are negligible ($\overline{\Delta S} \approx 0$). Since $\bar{P}$, $\overline{E_0}$ and $\bar{E}$ represent temporal averages, and we were also interested in temporal trajectories of those magnitudes, we computed time*

series of moving averages for each of the three variables. The temporal averaging window for which $\overline{\Delta S} \approx 0$ is typically unknown, however, it has been shown to exhibit a threshold behavior (i.e., above a certain averaging window size $\overline{\Delta S}$ is consistently negligible) (Han et al., 2020). The threshold averaging window size can vary between catchments, though approximately 71% of global catchments reach the threshold with an averaging window of 10 years while 94% of catchments reach the threshold with an averaging window of 30 years (Han et al., 2020). To address the uncertainty in the threshold averaging window size, we computed different "realizations" of the actual trajectories in terms of $\frac{\overline{E_0}}{\overline{P}}$ and $\frac{\bar{E}}{\overline{P}}$ for each catchment for all possible integer-year averaging windows in annual steps from one year to the full length of record. Then, the theoretical (or "conjectured") Budyko curve of Eq. (5) was fitted by adjusting the value of $n$ using the full length of record in each catchment.

The conjecture was tested for each reference catchment by statistically comparing all realizations of its actual trajectories to its theoretical Budyko curve trajectory using the non-parametric sign test (Holander and Wolfe, 1973). This is a distribution-free test for consistent over- or under-estimation between paired observations (see also Supplemental Information Sect. S2). If the catchment trajectory conjecture is correct, then the frequency at which actual and expected Budyko space trajectories are found to be statistically indistinguishable will be higher than what is expected due to random chance (see also Supplemental Information Sect. S2)."

2) Add a section to the supplementary information,

 Section S2.5 Controlling for potential catchment storage dynamics

"The temporal averaging window for which $\overline{\Delta S} \approx 0$ for the reference catchments in this study is unknown and may vary between catchments. However, it should be expected that above some threshold averaging window size, $\overline{\Delta S} \approx 0$ for many of the catchments most of the time (e.g., greater than a 10 year-average window), otherwise the reference catchments would rarely be in steady state. This threshold behaviour for $\overline{\Delta S} \approx 0$ has been shown to be near universal for catchments across Earth (Han et al., 2020), however, even if this was not the case for our reference catchments, and they are rarely in steady state, it should be expected that for some averaging windows for some catchments, steady state conditions are present (i.e., $\overline{\Delta S} \approx 0$). With or without this threshold behavior, testing all of the averaging windows for all catchments allows for a robust test of the catchment trajectory conjecture. In the case of threshold behavior, once the averaging window has reached sufficient size for a particular catchment (e.g., > 10 years), if the catchment trajectory conjecture is correct, the actual and expected Budyko space trajectories would be consistently statistically indistinguishable. This means that the frequency at which actual and expected Budyko space trajectories are found to be statistically indistinguishable would be higher than what would be expected due to random chance (i.e., more than 5% of all the possible actual and expected Budyko space trajectories would be statistically indistinguishable at a significance level of 0.05). An elevated frequency of statistical similarity would also occur if catchments were only rarely in steady state. The reason for this elevation is that the number of statistically similar trajectories from averaging windows where $\overline{\Delta S} \approx 0$ would be in addition to the number of statistically similar trajectories expected from random chance."

**Comment 4:**

**Might it be that water storage provides the link between the analytical meaning of the parameter and the physical understanding of how individual catchments have different E for the same P and Ep? That is, when a stored water term is included as a water supply term alongside P, then it introduces a catchment-specific and time-dependent term into the model fundamentals and into your analytical analysis?**

Response 4:

We agree that it might be possible that explicitly including storage changes in the water supply term (i.e., $\bar{E} = [\bar{P} - \overline{\Delta S}] - \bar{Q}$) would show that reference catchments' trajectories follow a particular parameter Budyko curve with a temporally constant catchment specific parameter (a "storage-dependent catchment trajectory conjecture"). However, to our knowledge, this hypothesis has not been explicitly tested within the literature. Additionally, the results from our empirical test (which implicitly incorporates storage dynamics) do not support this idea (see response to **Comment 3** above). Therefore, while it might be possible, the best currently available evidence does not appear to support such as hypothesis.

Furthermore, if the storage-dependent catchment trajectory conjecture is indeed correct, it can only be correct for one of the formulations of the parametric Budyko equations, since the various functional forms produce contradicting curves/trajectories when the catchment-specific parameter is held constant (see Sections 3.2.2 and 4.2.2 of the main text).

**References:**

Han, J., Yang, Y., Roderick, M. L., McVicar, T. R., Yang, D., Zhang, S., and Beck, H. E.: Assessing the Steady-State Assumption in Water Balance Calculation Across Global Catchments, Water Resources Research, 56, 10.1029/2020wr027392, 2020.
Holander, M., and Wolfe, D. A.: Nonparametric statistical methods, New York: John Wilew and Sons Inc. Publications, 497, 1973.

---

## Author Response (AR1)

**Editor's Comments (Received 20 April 2021)**

We thank the editor (Dr. Luis Samaniego) for his time and summary of the reviewers' comments, which we address in detail in the point-by-point responses below. Additionally, responses to the editor's comments are given below, with all line numbers corresponding to the revised manuscript unless otherwise noted. The following notation is used to refer to the numbering of reviewer comments, pages, and line numbers: RXCY = Reviewer X Comment Y; SC = Short Comment; AR = Author Response; PX:LY = Page X Line Y.

**Editor Comment 1:** **Two reviewers concluded that the MS need major revisions but that it has the potential to become a suitable contribution for the community. The main criticisms appear to be: is there a real need to reinterpret the B. framework? One Rev indicate that currently there are "several conceptual limitations that make the results only weakly support the main inferences of the paper" and the other pointing out: "the Budyko framework remains a powerful concept when interpreted and applied correctly. This Rev. doesn't think that a "reinterpret[ation]" the Budyko framework is a necessity.**

> AR: We appreciate the editor's summary of the reviewers' concerns and criticisms. The reviewers provided many thoughtful and useful comments, which we thoroughly address in our responses and revisions presented in the remainder of this document. We believe these changes significantly improved the manuscript over its original form and appreciate the opportunity to submit this revision.
>
> While the editor highlights several important concerns raised by the reviewers, we do not believe that the main criticism of the original manuscript was a question of the need for a reinterpretation of the Budyko framework. Both Reviewer 1 and 2 generally agree with the premise of the original manuscript (i.e., a reinterpretation of the framework). Specifically, Reviewer 1 states *"…it is my assessment that the line of arguments and conclusions, as presented in this paper, are mostly adequate"* and *"I largely agree with the general conclusions of the paper"* (**R1C2**). Additionally, Reviewer 2 states *"…the paper addresses two very relevant aspects of the Budyko framework…"* (**R2C2**) and *"…I think this paper could make an excellent contribution to the literature"* (**R2C5**).
>
> The reviewers' concerns highlighted in the editor's comment are largely related to the specific methodologies employed (Reviewer 2) and the scope of the reinterpretation (Reviewer 1) rather than the necessity of a reinterpretation. We thoroughly address the conceptual and methodological concerns of Reviewer 2 in our responses to **R1C9**, **R2C2-4**, **R2C17**, and **SCC3**. Reviewer 1 suggested that we do not reinterpret the entire Budyko framework, but rather only reinterpret explicit Budyko curves and parameters of the Budyko framework (**R1C3** and **R1C14**). We agree with this assessment and have revised the manuscript accordingly (addressing **R1C3**, as well as **R2C25** and **SCC2**). As an example of the scope of these revisions, we have changed the title of the manuscript to "Theoretical and empirical evidence against the Budyko catchment trajectory conjecture" based on **R1C14**.

**Editor Comment 2:** **Authors should carefully balance their opinions and interpretation, or based strong arguments against it, basin on factual findings using global data set. 728 reference catchments in the United Kingdom and United States represent only humid environments. With this sample no general statements can be draw. Please samples covering all climate regimes if you pursue radical ideas to Reinterpreting this Framework.**

AR: We believe that our careful revisions and responses thoroughly address all the points raised by reviewers and the interactive discussion. Our conclusions about the fundamental limitations of the parametric Budyko equations emerged from a genuine interest in the framework and a careful study of the catchment hydrology literature while attempting to improve the biophysical understanding of the catchment-specific parameters. The interpretations presented in the manuscript are wholly based on theoretical (**Sections 3.1.1 and 4.1.1**) and/or empirical evidence (**Sections 3.1.2 and 4.1.2**). This is in stark contrast to current commonly held interpretations of the parametric Budyko framework, which rely on untested assumptions and unsupported conjectures (see **P2:L29-P5:L3** and **Section 2.2**).

Importantly, the 728 reference catchments used here do not only represent humid environments as suggested by the Editor. This is evident in Figure 2 of the original manuscript, which plots the catchments in Budyko space. Of these 728 catchments, 300 are arid $\left(\frac{\overline{E_0}}{\overline{P}} > 1\right)$, and 428 are humid $\left(\frac{\overline{E_0}}{\overline{P}} < 1\right)$, collectively spanning a very wide range of aridity indices $\left(0.13 < \frac{\overline{E_0}}{\overline{P}} < 5.93\right)$. Additionally, they include 4 of the 5 main Köppen-Geiger (KG) Climate Classification Groups (arid, warm temperate, boreal, and polar) and 16 of the 31 KG sub-classifications. Given that the reference catchments used in this study span a wide range of aridities and climatic regimes, they allow us to draw robust and general conclusions about the Budyko framework from them, a point we have now highlighted in the revised manuscript (**P13:L19-31**):

"*The aridity indices of the 728 UK and US reference catchments span from 0.13 to 5.93 (300 are arid, $\phi > 1$, and 428 are humid, $\phi < 1$), and thus provide excellent coverage of Budyko space. Additionally, the reference catchments span a wide range of climates, capturing 4 of the 5 main Köppen-Geiger Climate Classification Groups (arid, warm temperate, boreal, and polar) and 16 of the 31 sub-classifications (hot desert, cold desert, hot semi-arid, cold semi-arid, humid subtropical, temperate oceanic, subpolar oceanic, hot-summer Mediterranean, warm-summer Mediterranean, cold-summer Mediterranean, hot-summer humid continental, warm-summer humid continental, subarctic, Mediterranean-influenced warm-summer humid continental, Mediterranean-influenced subarctic climate, and tundra) (Kottek et al., 2006;Rubel et al., 2017;McCurley Pisarello and Jawitz, 2020). With this broad and inclusive range of climatic conditions, robust and general conclusions about the Budyko framework and catchment trajectory conjecture can be drawn from this set of reference catchments.*"

**Anonymous Referee #1 ()**

We thank the reviewer for their time and helpful comments. Below are explanations of our responses to the reviewer's comments with all line numbers corresponding to the revised manuscript unless otherwise noted. The following notation is used to refer to the numbering of reviewer comments, pages, and line numbers: RXCY = Reviewer X Comment Y; SC = Short Comment; AR = Author Response; PX:LY = Page X Line Y.

**R1C1: The manuscript entitled "Reinterpreting the Budyko Framework" by Reaver and colleagues highlights several misconceptions regarding recent interpretations of results obtained using the Budyko framework. The authors especially criticize the common assumption that the wealth of functional Budyko curves represent expected trajectories through the Budyko space. By considering a stochastical model and observations from several hundred catchments, it is shown that catchment behavior in time does not follow the predicted trajectories within the Budyko space. The authors further highlight specific parameters used within parametric Budyko equations do not represent catchment-specific**

**biophysical features. The authors thus conclude that Budyko-based results should be interpreted more carefully and thoughtfully.**

> AR: We thank the reviewer for this accurate representation of the work, though we note that an additional outcome is to illustrate that the non-uniqueness of the parametric Budyko equations (i.e., there are several equally valid single parameter equations with different functional forms) fundamentally contradicts many recent interpretations of the parametric Budyko framework.

**R1C2: The manuscript is generally in good shape, overall well structured and well written. The introduction motivates the study and provides an in-depth overview of the recent research within the field. It needs to be acknowledged that this paper addresses a somewhat heated debate on the interpretation and applicability of the Budyko framework in the context of biophysical drivers influencing the terrestrial water and energy balance. However, it is my assessment that the line of arguments and conclusions, as presented in this paper, are mostly adequate. The supporting data and examples seem valid, but I would appreciate a more in-depth justification of several assumptions.**

> AR: We thank the reviewer for the kind words and the positive assessment of the manuscript. We address the request for additional justification of assumptions in subsequent responses.

**R1C3: I have provided a relatively small number of comments. However, I would also like the authors to consider two more general remarks:**

**(i) I largely agree with the general conclusions of the paper. However, I know (from my own experience) that the debate on the interpretability of the Budyko framework is somewhat heated. Therefore, I think it needs to be noted that the Budyko framework remains a powerful concept when interpreted and applied correctly. And I don't think that you necessarily "reinterpret" the Budyko framework. I fully agree that it needs to be acknowledged that there has been a rather large number of recent studies that overinterpreted results. Nonetheless, these studies still present results that are valid and sound within their specific setting. However, any interpretation going beyond these settings is not adequate, which needs to be acknowledged without condemning previous research. You already highlight this in your introduction, but I think you also need to be more careful throughout the rest of the manuscript (see some of the more specific comments below).**

**In this spirit, I would like to see a more positive evaluation of the Budyko framework per se. I think that the framework, given its adequate application and interpretation, remains super useful. It might thus help to better outline how upcoming Budyko-based research can profit from considering the limitations highlighted in this study. Your conclusions already provide some suggestions, but I still think that the Budyko framework has more potential besides being a global constraint (p. 25,l. 14), as it can also be applied within well-defined setups.**

> AR: We take the reviewer's point and have revised the manuscript to strengthen this concept (specific edits listed below) and to outline useful directions for future Budyko-based research (see also response to **R2C25**). In addition, we do not intend to condemn previous research and explicitly attempt to convey this sentiment in the manuscript (e.g., **P3:L20-24;** see also our responses to **R2C10 and SCC2**). Specifically, we acknowledge the need to be more specific and state that we believe that only the "original" Budyko framework (i.e., the observation that the aggregate behavior of multiple catchments consistently produce a distinctive pattern in Budyko space) remains intrinsically useful. Despite our critiques, we stress that the intent and much of the effort of previous research that utilized inappropriate interpretations of the parametric framework can be preserved (**P23:L9-14).**

To strengthen our support for the validity of the "original" Budyko framework in the manuscript, highlight how the intent and effort of previous research can be preserved, and better outline useful directions for future Budyko-based research, we have revised the manuscript as follows:

1) Added the following sentence to the introduction (**P3:L22-24**):
   *"Additionally, we emphasize that the Budyko framework based on the curve-like clustering pattern observed across multiple catchments is a powerful and useful concept when used appropriately and within the proper context"*

2) Added the following sentences to the end of Section 4.2.1 (**P24:L1-6**):
   *"While the acknowledgment of the proxy nature of the catchment-specific parameter and $\frac{\bar{E}}{\bar{P}}$ casts doubt on the specific conclusions of previous parametric Budyko-based research, we note that both the intent and much of the effort of many such studies can be preserved. For example, studies that related $n$ or $w$ to catchment biophysical features using various analytical tools could employ the same methods to relate $\bar{E}$ or $\frac{\bar{E}}{\bar{P}}$ to biophysical features directly. Doing so would preserve most of the analyses of such studies (i.e., near identical methods) as well as their intent (i.e., understanding the relationship between $\bar{E}$ and catchment biophysical features)."*

3) Substantially edited an existing paragraph in the conclusion section (**P28:L15-22**):
   *"Additionally, to be a valid representation of catchment evapotranspiration, process-based models need to able to reproduce the empirically established, nonparametric Budyko curve behavior when applied to multiple catchments across a range of climates. Thus, the general Budyko curve behavior can serve as a global constraint (i.e., calibration or validation) in the application of such models, e.g., Greve et al. (2020). Furthermore, while the parametric Budyko framework lacks predictive power, the nonparametric framework allows for probabilistic predictions of $\bar{E}$ and $\bar{Q}$ as well as changes in $\bar{E}$ and $\bar{Q}$ for ungauged basins. Within these contexts, the nonparametric Budyko framework is a tremendously useful conceptualization."*

**R1C4:** **(ii) Your theoretical example using Porporatos model is neat. However, it is still an artificial example and also needs to be interpreted as such. You use one model (Porporatos model) to investigate the characteristics of another model (Budyko). Fine, but you need to thoroughly justify that Porporatos model is an appropriate choice in this context: Is the choice of parameter values for the different cases realistic? What kind of conditions do these parameter values represent? Is there any real-world example that would illustrate your choice?**

**Additionally, as the parameters might be independent within your theoretical modeling framework, they might not be independent under real-world conditions. That also represents another problem of the large number of studies trying to identify biophysical controls. There is no single parameter that controls the partitioning of precipitation into evaporation and runoff. It is rather a convoluted mess of different processes that interact with each other.**

AR: Our primary justifications for using the Porporato model were its simplicity (**P10:L24-25**, **P11:L1-5**), physically-based nature (**P10:L11-13**), and its previous use within the Budyko literature (**P10:L9-11**). While we agree that the Porporato model is an artificial example (i.e., a model investigating a model) and has some limitations (e.g., assumes $E_0$ is constant), we do not believe that these properties limit the conclusions from the theoretical test. However, we take the

reviewer's point that the appropriateness of the Porporato model and the selected parameterization could be further justified in the manuscript and have done so by adding the following paragraph to the end of Section 3.1.1 (**P11:L20-23**):

*"The effective climate and landscape parameters in the Porporato model appear exclusively in ratios, such that only the relative magnitude between parameters is important. Moreover, the same trajectories can be made from infinite parameter combinations. For our test cases, we chose parameter values to maintain illustrative simplicity and to produce visually informative trajectories not restricted to a small portion of Budyko space."*

Additional justification for the choice of parameters values is described on **P11:L16-19.** We emphasize, however, that choosing values representing particular catchment conditions (e.g., an effective rooting depth of 0.374 m) would not change the results (since the same trajectories could be produced by adjusting other parameters), but the simplicity and clarity of the test cases would be lost. Finally, the reviewer rightly points out that there may be possible dependencies between model parameters and that the test cases should correspond to real-world examples. For the variable parameter test cases, we explicitly considered the possible dependencies of the climate parameters (i.e., the relationships between $\eta$ and $\psi$) (**P11:L16-19**) and chose test cases that are reflective of real-world conditions (**P11:L11-16)**.

**R1C5: p. 3, l. 7: Please also consider Padron et al., 2017. It provides a comprehensive overview of inconclusive and contradictory evidence obtained from using eq. 6.**

AR: We agree that Padrón et al. (2017) should be included and have added the following sentence (**P3:L6-9):**

*"Furthermore, Padrón et al. (2017) undertook a comprehensive overview of the wide variety of biophysical features proposed to control the catchment-specific parameter, finding that most proposed features did not actually correlate with the parameter and the types of features that were correlated varied substantially between climatic regions."*

**R1C6: p. 4, l. 4-7: It will be helpful to already mention those equations here.**

AR: Given the apparently contentious status of the topics covered into this manuscript (as the reviewer notes in **R1C2-3**), we have intentionally structured our introduction to ensure that the reader is informed of the full context, implications, and summary of the manuscript's contents before being introduced to the more detailed background, analyses, and discussions. We have kept this structure, however we understand the reviewer's point and therefore edited this sentence to (**P4:L16-18**):

*"We argue and demonstrate herein that the two widely accepted parametric Budyko equations (i.e., those derived in Zhang et al. (2004) and Yang et al. (2008)) are non-unique, meaning they are only two of many possible single-parameter Budyko equations."*

**R1C7: Sec. 2.3: I think it might help to incorporate this section into Sec. 2.1?**

AR: We understand that "Budyko's interpretation of explicit curves" could naturally be classified as part of an "Overview of the Budyko hypothesis and equations", however, we specifically placed it after "Current interpretations of explicit Budyko curves and the parametric framework" to contrast how current interpretations have evolved (or strayed) from the "original" intent of explicit curves. This section provides the start of our reinterpretation (or "retrospective interpretation") from the current state of the framework. Therefore, we believe it is important to

highlight Budko's interpretation independently and following the introduction of current interpretations.

**R1C8: p. 10, l. 14: It might be helpful to further explain what you mean by "Budyko-like"?**

AR: We revised this sentence to (**P10:L17**):

*"We first write the model of Porporato et al. (2004) in a form that can be plotted in Budyko space,"*

**R1C9: p. 10, eq. 8: Could you explain in more detail how you estimate the aridity index from this equation?**

AR: We added the following sentence to make this clearer (**P11:L5-6**):

*"The ratio of $\psi$ and $\eta$ is the aridity index, $\phi = \frac{\overline{E_0}}{\overline{P}} = \frac{\overline{E_0}}{\alpha\lambda} = \frac{\psi}{\eta}$."*

**R1C10: p. 11, l. 15: Why 2m? Well, this is related to my second major comment above. The choice of these parameter values needs to be justified. What kind of soil characteristics does Z0=2m represent? I know that your overall conclusions won't change when setting Z0=1.9, but it is important to understand what it means and what kind of real-world characteristics your choice represents.**

AR: We have addressed this comment in our response to **R1C4**.

**R1C11: Sec. 3.2: Why don't you include this subsection in the Background part (Sec. 2.)?**

AR: The Background section currently only contains existing information in the literature. We believe the content of **Section 3.2** is a new contribution, and therefore we placed the content in the methodological and discussion portions of the manuscript.

**R1C 12: p. 14, l. 8-10: Is that true? Are Eqs. 5 and 6 considered the only valid parametric Budyko equations? Do you have more evidence for this statement?**

AR: We thank the reviewer for addressing this point. Upon review, we agree that there are other single-parameter functional forms that also satisfy the uniqueness requirement. Based on this comment and **R2C20**, we have substantially revised **Section 3.2.2** to focus on the properties that are typically used to justify the validity of Eq. (5) and (6) and highlight how these properties are not unique to these equations. See detailed manuscript changes in our response to **R2C20.**

**R1C13: Figure 2: I know it is hard to convey all the necessary information into one Figure, but I have to admit that this one is especially difficult to interpret. The trajectories are a big mess (and to a certain extent this is exactly what you want to highlight here). However, Figure 3a is of more value in this context. If you like to keep Figure 2, maybe consider drawing thinner red lines or introduce some transparency?**

AR: We agree with these suggestions and those provided in **R2C23**. We reduced the thickness of the red lines, logarithmically (base 10) scaled the abscissa, and modified the text referencing this figure accordingly.

**R1C14: p. 25, l. 1-2: Maybe this would be a better title: A reinterpretation of explicit Budyko curves and parametric Budyko equations.**

AR: The original title of the manuscript lacked sufficient specificity. Therefore, we adopted a title that provides a more focused representation of the contents of the manuscript: *"Theoretical and empirical evidence against the Budyko catchment trajectory conjecture"*

**R1C15:** p. 25, l. 20-26: I agree that the interpretation of the parameter representing landscape features is misleading. Calling it a catchment-specific parameter is not justified either. However, even though it is a lumped parameter just existing as a mathematical necessity without any a priori physical interpretation, there might still be an a posteriori physical interpretation. You call the parameter a proxy variable for E/P, which is, in fact, also some sort of physical interpretation. That means, if you assume a constant aridity index and change E/P, the parameter changes as well. Vice versa, if you change the parameter, E/P changes as well. I think the misleading interpretation here is often more related to the assumption that the parameter somehow controls E/P, which is definitely not true.

> AR: As we state in the manuscript (**P14:L15-29**), in order for the "catchment-specific parameter" to have *a posteriori* physical interpretation, the associated functional form (i.e., Eqs. (5) or (6)) would have to be empirically valid. However, since our empirical test of the catchment trajectory conjecture refutes the empirical validity of parametric Budyko curves, making Eqs. (5) and (6) under-determined, we conclude (**P22:L17-30**) that the "catchment-specific parameter" does not have an interpretation independent of $\frac{\overline{E_0}}{\overline{P}}$ and $\frac{\overline{E}}{\overline{P}}$.
>
> We say the "catchment-specific parameter" is a proxy for $\frac{\overline{E}}{\overline{P}}$ since, in practice, the values of $\frac{\overline{E_0}}{\overline{P}}$ (usually taken as a known quantity) and $\frac{\overline{E}}{\overline{P}}$ *always* determine the "catchment-specific parameter". For real catchments, the association between $\frac{\overline{E}}{\overline{P}}$ and the "catchment-specific parameter" is one-way; $\frac{\overline{E_0}}{\overline{P}}$ and $\frac{\overline{E}}{\overline{P}}$ are used to compute $n$ or $w$. It is never the case that $\frac{\overline{E_0}}{\overline{P}}$ and $n$ or $w$ are used to compute $\frac{\overline{E}}{\overline{P}}$. Therefore, we agree with the reviewer that the "catchment-specific parameter" does not control $\frac{\overline{E}}{\overline{P}}$, however $\frac{\overline{E}}{\overline{P}}$ completely controls the value of the "catchment-specific parameter" (see **P23:L1-7**).

**R1C16:** p. 25, l. 27-28: This statement is too strong in my opinion (see also my first major comment). Any interpretation of obtained results is valid within their specific setting. However, it is the overinterpretation and generalization that is "untenable" (which is a very strong word in this context).

> AR: We have largely addressed the philosophical underpinnings of this comment in our response to **R1C3**. We take the reviewer's point about the strong wording of this statement and have revised the sentences in question (**P29:L2-5**)
>
> *"The collective results from our analyses suggest that current commonly held interpretations of Budyko curve trajectories and the parametric Budyko equations are unsupported. We propose that the catchment hydrology community look critically at the well-accepted but unjustified interpretations that are the current commonly held standard."*

**Anonymous Referee #2 ()**

We thank the reviewer for their time and helpful comments. Below are explanations of our responses to the reviewer's comments with all line numbers corresponding to the revised manuscript unless otherwise noted. The following notation is used to refer to the numbering of reviewer comments, pages, and line numbers: RXCY = Reviewer X Comment Y; SC = Short Comment; AR = Author Response; PX:LY = Page X Line Y.

**R2C1:** This paper presents interesting tests and perspectives on the Budyko framework. It first argues that there is no theoretical or empirical basis for typical key assumptions in the use of the framework

**(i.e (i) catchments follow parametric Budyko curves under aridity change, and (ii) the catchment-specific parameter (i.e. n or w) is determined by catchment biophysical properties. Subsequently, the paper aims to test these assumptions using outcomes of the Porporato 2004 model, and empirical data.**

AR: We appreciate the reviewer's interest the topics covered in our manuscript. The description provided is accurate, however as noted in our response to **R1C1**, another outcome of this work is to illustrate that the non-uniqueness of the parametric Budyko equations fundamentally contradicts many recent interpretations of the parametric Budyko framework.

**R2C2: While the paper addresses two very relevant aspects of the Budyko framework, there seem to be several conceptual limitations that make the results only weakly support the main inferences of the paper, because:**

**\* The approach using the models is that: "We tested the catchment trajectory conjecture by varying the model climatic parameters while holding the landscape parameter constant. If the resulting trajectories are not Budyko curves, the conjecture should be rejected." This approach assumes that the only relevant climate variable is aridity, but in reality as earlier work has shown (and as the model shows) other climate variables (such as seasonal cycles and P intermittency) also strongly controls water balances. Thus, the observation that the model diverges from the Budyko curves only shows that climate also matters (as is already known) and does not show that catchments do not follow the Budyko trajectory conjecture.**

AR: We thank the reviewer for acknowledging the relevance of the topic, however it is incorrect to state that our theoretical approach assumes that aridity is the only relevant climate variable. Clearly, many different climate properties control the water balance, and our theoretical approach did not seek to test how climatic properties affect the water balance (which is already well described). Rather, we tested the commonly accepted catchment trajectory conjecture, which states that individual catchments undergoing changes in aridity index will follow explicit Budyko curve trajectories.

We acknowledge, however, that our description of the catchment trajectory conjecture, Porporato model, and model variables can be improved. Revisions presented in our response to **R1C9** and those listed below aim to represent these conceptualizations better in the main text:

1) Revised sentences in the introduction (**P3:L25-31** and **P4:L1-7**):

*"We first re-examine interpretations of Budyko curves that ascribe physical meaning to the functional form of the curve, thus implying that explicit curves govern catchment evapotranspiration (e.g., Wang et al., 2016a;Wang and Hejazi, 2011;Jiang et al., 2015;Liang et al., 2015;Jaramillo et al., 2018;Zhang et al., 2004;Zhang et al., 2018). This concept is typically articulated through the suggestion that an individual catchment undergoing only changes in aridity index will follow an explicit Budyko curve trajectory ("the catchment trajectory conjecture"). However, we note that it is mathematically impossible for the aridity index to vary independently of other climate variables that impact $\overline{E_0}$ or $\overline{P}$, meaning that the catchment trajectory conjecture, as typically stated, is ill-posed and untestable. Given the stated conjecture's mathematical impossibility, in practice, it is generalized implicitly (or unintentionally) to a well-posed and testable form that suggests individual catchments with stable basin characteristics that undergo changes in aridity index will follow an explicit Budyko curve trajectory. Here we examine the support for the well-posed conjecture and test it, the results of which suggest that specific functional forms of Budyko curves do not have intrinsic physical meaning, but are instead semi-*

*empirical conceptual tools that describe the general aggregate behavior of multiple catchments—but do not predict the specific behavior of individual catchments."*

2) Revised a sentence in Section 4.1.1 (**P16:L19-20**):

*"The main conclusion of this theoretical test is that a catchment undergoing changes in aridity index does not have to follow a Budyko curve, contrary to the catchment trajectory conjecture."*

While we hope that these revisions improve the manuscript's clarity, we maintain that the results and conclusions from the current theoretical tests are sound and posit that the reviewer's comment may be due to discrepancies in conceptualization of the aridity index and catchment trajectory conjecture, which we seek to clarify in the remainder of our response to this comment.

First, as we state in the manuscript (**P3:L28-29**), the catchment trajectory conjecture suggests that individual catchments undergoing ***only*** changes in aridity index will follow an explicit Budyko curve trajectory. While this is the common interpretation (as the reviewer suggests), it is an ill-posed conjecture since it is not possible to ***only*** change the aridity index independent of other climatic variables. Therefore, it does not make sense to conceptualize $\phi$ as an independent or isolated climate variable, even though this is often done in the Budyko framework.

Second, the catchment trajectory conjecture suggests that catchments should follow specific parametric curves when undergoing changes in $\phi$, but it doesn't specify the mechanism for how $\phi$ changes (e.g., changes in precipitation properties vs changes in potential evaporation properties). Different mechanisms of $\phi$ change will impact the evaporative index in different ways, almost all of which lead to catchments not following the particular parametric or non-parametric curves, as illustrated by the theoretical tests we present in the manuscript. This implies that individual catchments typically **do not** follow Budyko curve trajectories.

Finally, the Porporato model illustrates the dependency of $\phi$ on "other" climate variables explicitly (even with its simplified dynamics). Within the model, $\phi = \frac{\psi}{\eta} = \frac{\overline{E_0}}{\overline{P}} = \frac{\overline{E_0}}{\alpha\lambda}$. To change the value of $\phi$, one must change either the average storm depth, $\alpha$, the average storm frequency, $\lambda$, or the average potential evaporation, $\overline{E_0}$. Changing one or another variable produces markedly different trajectories in Budyko space (see **Section 4.1.1**). We note that for real catchments, there are vastly more ways for $\phi$ to change than the simple Porporato model allows (since it only has three climate parameters). As such, it is even *less likely* for real catchments to follow specific Budyko curve trajectories; a statement which is supported by the results of our empirical test.

**R2C3: * The approach using the data: It is stated that "this prediction can be tested by comparing actual Budyko space trajectories of reference catchments computed from empirical observations against the expectation from the catchment trajectory conjecture. If the observed reference catchment trajectories are distinct from the expected Budyko curve trajectories, the conjecture should be rejected.". However, there are many other reasons why the trajectories do not follow the Budyko curves. For example, the water balance may not be closed, measurements may be off, climate variables (other than aridity) may also change (since land-cover is the only variable which is controlled for). Therefore it seems somewhat unfair to attribute any anomaly from the curve to solely the Budyko trajectory being wrong, rather than that is also could be caused by any of the other factors.**

AR: We agree that there are many reasons why reference catchments may not follow Budyko curve trajectories over time. Part of our motivation for testing the catchment trajectory conjecture was that the widespread default assumptions about catchments' long-term

hydrological behavior was based on an untested and ill-posed assertion (see also response to **R2C2**). As the reviewer points out, certain conditions (e.g., closed water balance) must be accounted or controlled for when conducting a rigorous test of the catchment trajectory conjecture. The reviewer's first example about potential issues with the closure of the water balance and its implications for estimating $\bar{E}$ via $\bar{P} - \bar{Q}$ was also raised in **SCC1**. Our methodology for the empirical test (**P12:L7-30**, **P13:L1-2**, and **Section S2 of the Supplemental Information**) was specifically chosen to robustly address catchment water balance closure and storage dynamics that can impact the estimation of $\bar{E}$. Therefore, our results and conclusions from the test (i.e., a rejection of catchment trajectory conjecture) are robust. We detail these specific arguments in our response to **SCC3**, where we also describe edits to the manuscript to make this point clearer.

The reviewer's second example of potential misattribution concerns error in the empirical measurement of $E_0$, $P$, and $Q$. We agree that this is possible for hydrological and meteorological data in any hydro-climatological study, however the data used in this study are from peer-reviewed datasets and were produced using standardized methodologies by the governments of the US (e.g., USGS, NASA, and NOAA) and UK (e.g., NRFA and Met Office). If the errors present in these data are sufficient to obscure catchments' "true" Budyko curve trajectories, it is unlikely that any current continental-scale catchment datasets would have sufficient accuracy to apply the parametric Budyko equations in a meaningful way. Given that individual catchments have not been shown to follow specific Budyko curve trajectories previously (see **P8:L5-27**), and our study reaffirms this finding for 95% of realizations over 728 rigorously measured catchments (see **P19:L19-25**), we contend that our findings are robust. To highlight this point, we revised a sentence in Section 3.1.2 (**P13:L3-5**):

*"Our empirical tests were based on 728 UK and US reference catchments identified from well-accepted peer-reviewed datasets. These datasets were produced using standardized methodologies with well-documented quality control standards."*

The final example the reviewer gives as a possible cause of non-Budyko curve behavior is that climate variables other than aridity could be changing and were not controlled for. We have addressed this comment in our response to **R2C2**.

**R2C4: * To what extent are the methods sensitive to the use of the Hargreaves potential evaporation over any other PET estimate? In theory it seems that the ambiguity of the PET estimate has similar problems as that of catchment specific parameter in the Budyko framework (e.g. suffering from non-uniqueness and potentially crossing trajectories).**

AR: We confirmed that the conclusions obtained from our empirical test methodology are insensitive to the choice of $E_0$ method (e.g., Hargreaves vs. others) and added a new **Section S2.6** to the Supplementary Information describing this analysis. Therefore, the analyses presented are a robust test of the catchment trajectory conjecture. To highlight this point in the manuscript, we revised a sentence in Section 3.1.2 (**P13:L15-18**):

*"Daily $E_0$ time series were computed from the daily $T_{max}$ and $T_{min}$ values using the Hargreaves potential evaporation equation (Hargreaves and Allen, 2003;Lu et al., 2005;Allen et al., 1998), though we note our empirical test methodology is insensitive to the specific choice of $E_0$ method (see Supplemental Information Sect. S2.6)."*

In short, there are four lines of support for our conclusion that the empirical test methodology is insensitive to the choice of $E_0$ method: (1) empirical evidence of insensitivity - we used two different $E_0$ methods in the test of the catchment trajectory conjecture (Hargreaves for US

catchments and Penman-Monteith for UK catchments), and both produced the same conclusion: rejection of the conjecture (**P13:L8-18** and **P19:L20-25**); (2) the various possible $E_0$ methods that could be used in this analysis are highly correlated, so the choice of method will not alter the basic shape of the actual Budyko space trajectories (see **Section S2.6** and **Figure R1**); (3) for all averaging periods, catchments' trajectories are overwhelmingly driven by changes in $\bar{P}$ rather than changes in $\bar{E_0}$ (see **Section S2.6** and **Figure R2**); and (4) the non-parametric sign test used to determine consistent differences between actual and expected trajectories will provide near-identical results for each possible $E_0$ method if the basic shape of the actual Budyko space trajectories are generally preserved (see **Section S2.6 and Figure R1)**.

[Figure]

Figure R1: Illustration of the potential effects of the choice of $E_0$ method on the empirical test of the catchment trajectory conjecture. The actual (red solid curves) and expected trajectories (black dashed curves) of a catchment calculated using one particular potential evapotranspiration method, $E_0^1$, are given on the left side of the figure. If a different method, $E_0^2$, gives estimates of $E_0$ twice that of $E_0^1$, (i.e., $E_0^2 = 2E_0^1$), then the catchment's actual trajectory translates along the $\phi$ axis and symmetrically expands around its average value in the $\phi$ dimension. The expected trajectory changes to a new parametric Budyko curve. This new expected trajectory for $E_0^2$ will be slightly rotated with respect to actual trajectory as compared to the trajectories computed with $E_0^1$. However, the relative frequency of over- and under-estimation of the expected trajectory compared to the actual trajectory remains essentially unchanged. Thus, outcome of the non-parametric sign test is the same for both $E_0$ methods.

[Figure]

Figure R2: Corresponding standard deviations of $\overline{E_0}$ and $\overline{P}$ for all possible actual Budyko space trajectory realizations used in the empirical test of the catchment trajectory conjecture (gray dots). Histograms of the marginal distributions for $\overline{E_0}$ and $\overline{P}$ realizations are shown in red and blue, respectively. Nearly all points fall below the 1 to 1 line (black dashed line) meaning changes in $\overline{P}$ dominate the temporal dynamics of catchments' trajectories. The mean standard deviation of $\overline{P}$ is 0.22 mm/day (blue dashed line), 5.5 times larger than the value for $\overline{E_0}$, 0.04 mm/day (red dashed line).

**R2C5: Beside these limitations, the paper nicely contrasts the large number of of Budyko studies that "blindly apply Budyko equations", and emphasizes some shortcomings of the framework that are too often ignored. Once the above issues are addressed (and the detailed comments below) I think this paper could make an excellent contribution to the literature.**

AR: We thank the reviewer for the positive evaluation of the manuscript's contribution to the literature. We believe that we have addressed the reviewers concerns in our responses to the comments and with our suggested edits to the manuscript.

**R2C6: Page 1 L18: "components" or "assumptions"?**

AR: We agree that "assumptions" is a more accurate word and changed this wording throughout the manuscript.

**R2C7: Page 2 L3: "rainfall" should be changed into "precipitation" (as it also includes snow). This change is also recommended at other places where rainfall is stated, but precipitation may be more appropriate.**

> AR: We agree with the reviewer and changed the wording at all places where "rainfall" was used inappropriately in the manuscript.

**R2C8: Page 2 L30: Note that the Gentine et al. (2012) study excluded most US catchments with loads of snow or out of phase precipitation regimes (i.e. Mediterranean). As a consequence, most scatter was removed, resulting in this interpretation.**

> AR: We thank the reviewer for calling this to our attention. Based on our reading Gentine et al. (2012), we do not believe their interpretation was completely dependent on the amount of scatter removed in their methodology. However, while they did not explicitly exclude Mediterranean and snowy climates, their exclusion methodology was biased against these types of catchments within the MOPEX dataset. Therefore, we retained the citation but also added context to their interpretation by editing the sentence in question to (**P3:L1-4):**

> *"For example, Gentine et al. (2012) suggested that the aggregate Budyko curve behavior already reflects the interdependence among vegetation, soil, and climate, and therefore, the inclusion of catchment-specific parameter into the Budyko framework is unnecessary. However, this interpretation was partially based on catchment data with limited scatter in Budyko space."*

**R2C9: Page 3 L13: It is stated that "we critically reinterpret two key and interrelated components of the current framework:". I am unsure these two things can be called "components". They are rather typical assumptions that people make, but as past authors acknowledge (as referenced by this paper, or as stated above in this review) these assumptions appear largely unfounded, untested, or premature.**

> AR: We agree that "assumptions" is a more accurate word and changed this wording throughout the manuscript (also see response to **R2C6**).

**R2C10: Page 3 L15-17: I appreciate the paper is trying to be gentle towards past research by saying "However, we stress that the aim of this reinterpretation is not to discard the voluminous efforts put forth using current interpretations of the Budyko framework, but rather to recontextualize the conclusions obtained from them". However, your work suggests that all attributions and sensitivity applications will have substantially wrong numbers. This obviously is important "context" but I'd rather say they also cast doubt on many of the past conclusions.**

> AR: We thank the reviewer for recognizing our attempt to treat previous work fairly. Additionally, we agree that our results do cast doubt on some of the conclusions of previous work (e.g., causal attributions and sensitivity applications). However, we also stress that both the intent and much of the effort of previous work can be maintained with an appropriate interpretation of the parametric Budyko framework. For example any study that has related $n$ or $w$ to catchment biophysical features can easily drop the parametric framework from their analysis and use their same analytical tools to relate $\bar{E}$ or $\frac{\bar{E}}{\bar{P}}$ to biophysical features directly (see **P23:L9-14**). This would preserve most of the analyses of such studies (i.e., same analytical methods) as well as the intent (i.e., understanding the interactions between $\bar{E}$ and catchment biophysical features). We attempt to improve our treatment of the recontextualization of prior work in the edits provided in response to **R1C3**.

**R2C11: Page 5 L14: Schreiber, 1904 was not aware yet of the concept of potential evapotranspiration, so I am unsure it is appropriate to cite this work here.**

AR: We have revisited the original German text of Schreiber (1904). We agree that he did not specifically use the concept of potential evapotranspiration, however, he did use a functionally equivalent constant "k" in its place (see explanation below), and he also seems to be the first to propose a functional form of what we now call the Budyko equations. Additionally, others, such as Ol'Dekop (1911), used and expanded upon his work to improve our understanding of the catchment water balance. Therefore, we think it is appropriate that his work is cited in reference to early work on the catchment water balance.

Schreiber (1904) constant "k":

Schreiber (1904) defines "k" as the limiting value that the difference between mean annual precipitation and runoff ($\bar{P} - \bar{Q}$, referred to as "die Rückstandshöhe" or the catchment's residue/hold-back height) approaches as precipitation becomes large (i.e., $\bar{P} \to \infty$). Quoting the specific passage:

*Je größer x [der jährichen Niederschlagschöhe] wird, um so kleiner wird $\frac{k}{x}$, so daß man für sehr große x*

$$y = x - k$$

*[die jähriche Abflußhöhe] setzen kann. Heiraus ergibt sich sofort die physikalische Bedeutung des Exponenten k als die Größe, der sich die Differenz zwischen Niederschlag und Abfluß [y] um so mehr nähert, je größer der Niederschlag selbst wird. Dieses Verhältnis scheint mir in der Natur des Problemes begründet zu sein. Die Differenz*

$$z = x - y$$

*kann man als die Rückstandshöhe bezeichen.* Schreiber (1904), page 3.

In our current language, the constant k would be the mean annual value of evapotranspiration under energy-limited conditions, i.e., the mean annual potential evapotranspiration, $\overline{E_0}$. However, while constant k is functionally equivalent to $\overline{E_0}$, Schreiber (1904) does not discuss or specify how the water that does not become discharge is being "held back" (i.e., does not discuss it as being evaporated) and therefore does not explicitly introduce the concept of potential evapotranspiration. Subsequent investigations by Ol'Dekop (1911) ascribed the concept of maximum possible evaporation (i.e., potential evapotranspiration) to "k", as detailed in Andréassian et al. (2016).

**R2C12: Page 5 L18: identical $\bar{P}$ and $\overline{E_0}$ seems somewhat inaccurate, as it is about the ratio of the two.**

AR: We agree that this statement is incomplete and have changed the sentence to (**P6:L1-2):**

*"However, Eq. (3) and other forms of Eq. (2) are unable to explain differences in $\bar{E}$ or the evaporative ratio between catchments with identical $\bar{P}$ and $\overline{E_0}$ or aridity indices, respectively."*

**R2C13: Methods. Page 10 L6: "(e.g., (" the second layer of brackets seems redundant**

AR: We removed the second layer of parentheses throughout the manuscript.

**R2C14:** Page 11 Section 3.1.1: This test seems inappropriate for its cause, because climate characteristics other than aridity are varied. (Also, see main comment above).

AR: We have addressed this concern in our responses to **R2C2-3**. The key idea from these responses is that it is mathematically impossible to only change the aridity index independent of other climatic characteristics.

**R2C15:** Page 11 Section 3.1.2: Note that similar types of test have been done in https://onlinelibrary.wiley.com/doi/full/10.1002/hyp.9949 and https://www.nature.com/articles/ncomms11603

AR: We thank the reviewer for highlighting these two additional references. We agree that the nature of the test conducted by Berghuijs and Woods (2016) is comparable to our approach (though it doesn't explicitly control for land use stability, i.e., using only reference catchments). We acknowledge Berghuijs and Woods (2016) in our edits in response to **R2C22**. While van der Velde et al. (2014) do track temporally changing Budyko space trajectories for individual catchments using a method somewhat similar to our approach, they do not focus on the behavior of individual catchments. Rather, they use trajectories to understand how groups of catchments have behaved and how they might do so in the future. Thus, we acknowledge van der Velde et al. (2014) in reference to the methodologies of calculating Budyko space trajectories. Specifically, we changed the sentence to (**P12:L9-11**):

*"Since $\bar{P}$, $\overline{E_0}$ and $\bar{E}$ represent temporal averages, and we were also interested in temporal trajectories of those magnitudes, we computed time series of moving averages for each of the three variables, similar to the method employed by van der Velde et al. (2014)."*

**R2C16:** Page 11 L18-20: selecting catchments with stable land-use makes the assumption that all other time-varying factors controlling the catchment's water balance (besides aridity) are irrelevant, but this is inaccurate as, for example, seasonal cycles of P can strongly vary between years (and strongly influence the precipitation partitioning).

Page 11 L20: as a consequence, it is hard to agree with "must be attributed to climatic factors and the catchment trajectory conjecture predicts that their expected trajectories through Budyko space must be Budyko curves". Are the ways to address this critical limitation (given its purpose) to your test?

AR: We agree that other climatic factors (e.g., varying seasonal cycles of $P$) can strongly impact a catchment's water balance. We have addressed the points brought up in this comment in our responses to **R2C2, R2C4,** and **R2C14.** The key idea is that it is mathematically impossible to only change the aridity index independent of other climatic characteristics, making the typical catchment trajectory conjecture ill-posed. In its original form, the conjecture is untestable since it does not specify the mechanism by which the aridity index should change over time (e.g., varying seasonal cycles of $P$) to produce specific parametric Budyko curve trajectories. We believe we have clarified these points and contend that our empirical test provides a well-formed and robust test of the catchment trajectory conjecture.

**R2C17:** Page 11 L27: "$\bar{E}$ were calculated from the catchment water balance, $\bar{E} = \bar{P} - \bar{Q}$." is an obvious way to approach the problem, but also known to have issues: https://agupubs.onlinelibrary.wiley.com/doi/10.1029/2020WR027392. What are the potential effects of storage changes (even over 10-y time-scales).

AR: We thank the reviewer for highlighting Han et al. (2020), as their work supports the validity of our empirical test methodology as described in responses to **R2C3** and **SCC3**. Specifically, if $\overline{\Delta S} \sim 0$ for some of the references catchments used for some averaging windows, then the results of our empirical methodology provide a robust test of the validity of the catchment trajectory conjecture. Han et al. (2020) found that under their tightest restrictions, 71% of the 1057 catchments they tested had $\overline{\Delta S} \sim 0$ for an averaging window of 10 years. Furthermore, 94% of their tested catchments had $\overline{\Delta S} \sim 0$ for an averaging window of 30 years. We note that our empirical methodology tested actual and expected trajectory realizations for all 728 reference catchments for all possible averaging windows, ranging from 1 to 45 years (see **P12:L7-19** and **P13:L9-18**). Therefore, based on the results of Han et al. (2020), we would expect nearly all of the reference catchments used to have $\overline{\Delta S} \sim 0$ for at least one of the averaging windows used, with the significant majority of catchments having $\overline{\Delta S} \sim 0$ for many of the averaging windows. We acknowledge the work of Han et al. (2020) with the edits described in our response to **SCC3**.

**R2C18:** Page 12: "applying moving-average window sizes ranging in annual steps from 1 year to the full length of record." How is it justified to use 1-year windows as these clearly can violate the delta S =~0 assumption?

AR: The conclusions from our empirical methodology leverage expected trajectory realizations for all 728 reference catchments for all possible averaging windows and are unaffected by whether certain catchments under certain averaging windows violate steady state conditions (see responses to **R2C3**, **R2C17**, and **SCC3**).

**R2C19:** Page 12 L25: Why Hargreaves PET, and are there any changes to the results when other PET estimates would be used?

AR: We have addressed this comment in our response to **R2C4**.

**Reviewer 2 Comment 20:**

**Page 14: It remains unclear to me what the purpose is of section 3.2.2. (Yes I see WHAT is done, but it seems not really explained WHY this is done).**

AR: The purpose of Section 3.2.2 (and Section 4.2.2) is to illustrate that single-parameter Budyko equations are non-unique, making the various different functional forms contradictory under commonly held interpretations of the framework. While this motivation was introduced in the Abstract (**P1:L26-27**) and Introduction (**P4:L16-19**), we agree that it could be better contextualized and motivated. We do so by substantially adding to and editing **Section 3.2.2** (**P15:L7** through **P16:L7**).

**R2C21:** Results and Discussion. Section 4.1.1. All these results seem to show that climate variables other than aridity also affect the partitioning of P into Q and E. This seems to be a strange way to test the catchment trajectory conjecture because if the resulting trajectories are not Budyko curves, it just means that climate (other than aridity) also influences the water balances, rather than being a test of the catchment trajectory conjecture. (See main comments).

AR: We have addressed this comment in our responses to **R2C2, R2C3**, **R2C14**, and **R2C16**. We reiterate the following points: 1) The aridity index cannot vary independent of other climate features, and 2) The catchment trajectory conjecture makes no claim about which climate properties vary to vary the aridity and evaporation indices in a way which will produce a specific Budyko curve trajectory (climate properties controlling $\overline{E_0}$ or $\overline{P}$ or both?). Because of points 1 and

2, the catchment trajectory conjecture is ill-posed. Despite the ill-posed and previously untested nature of the catchment trajectory conjecture, we believe that this work provides a fair assessment of its validity.

**R2C22:** **Section 4.1.2. L6: "their global behaviour". Can this be made more specific (i.e. does it refer only to the long-term mean water balances (e.g. black markers)?). Questioning that the prevailing interpretations of Budyko curves suggest that the explicit functional forms represent trajectories through Budyko space for individual catchments undergoing changes in aridity index has also been discussed in https://onlinelibrary.wiley.com/doi/10.1002/hyp.13958 and tested in https://www.nature.com/articles/ncomms11603. This may be worth acknowledging.**

AR: Yes, by global behavior we mean the long-term mean water balance (the black markers in Figure 2). To make this clearer we revised a sentence in this section to (**P19:L4-6**):

*"The catchments investigated span a wide range of aridity indices, climate zones, latitudes, longitudes, and vegetation types, and the global behavior of their long-term mean water balances is in agreement with the non-parametric Budyko curve (Fig. 2)"*

We thank the reviewer for highlighting the two additional references in relation to our empirical test of the catchment trajectory conjecture. We also agree that Berghuijs et al. (2020) and Berghuijs and Woods (2016) should be acknowledged. Thus, we made the following revisions:

1) Change sentences to (**P8:L24-27**):

   *"Additionally, interpretations of these relationships implicitly assume that the functional forms of either Eq. (5) or Eq. (6) represent a physically meaningful relationship between the aridity and evaporative indices, an assumption which has not been empirically validated, as previously noted by Berghuijs et al. (2020)."*

2) Change sentences to (**P20:L3-9**):

   *"The full range of evaporative index errors spanned from 0.4% to 1991%, with a mean of 26%. The mean error closely agrees with the value (27.9%) found by Berghuijs and Woods (2016) in a comparable test of the catchment trajectory conjecture using Eq. (6) and 420 catchments from the MOPEX dataset (Schaake et al., 2006). Importantly, the average relative error for the parametric Budyko framework (26%) is actually larger than that for Eq. (3) (23%), which suggests that the non-parametric Budyko curve is in better agreement with the global behavior of catchments than the ensemble of parametric curves specifically fit to the individual catchments."*

**R2C23: Figure 2: is there any way to better visualise what is going on here? One minor change (that will not resolve all issues) may be to make the x-axis on a log-scale. This avoids that humid catchments are all condensed in a tiny part of the left side of the figure (whereas arid catchments are spread out at the right-hand side).**

AR: We believe the edits detailed in our response to **R1C13** address these issues.

**R2C24: Section 4.2.2. L7-8: please specify that it is the common interpretation not ALL interpretations in "should cast doubt on the current interpretations of parametric Budyko equations,"**

AR: We agree we should be clearer on this point. We changed a sentence to (**P24:L13-15**):

*"The contradiction between Eq. (5) and (6) alone should cast doubt on current commonly held interpretations of parametric Budyko equations, particularly regarding the physical meaning of explicit curves and the provenance and meaning of the catchment-specific parameter."*

**R2C25:** **Conclusions. "We suggest that process-based evapotranspiration models be used" Note that this is consistent with earlier works: e.g. https://doi.org/10.1029/94WR00586, https://agupubs.onlinelibrary.wiley.com/doi/full/10.1029/2005WR004606, etc).**

AR: We agree that our suggestion has been recognized and implemented many times in previous work, so we revised a sentence to (**P28:L12-15**):

*"Therefore, as an alternative to using explicit Budyko curves to understand catchment trajectories, we re-iterate the long-standing suggestion (e.g., Eagleson (1978);Milly (1994);Daly and Porporato (2006);Rodriguez-Iturbe et al. (1999);Feng et al. (2015), etc.) that process-based evapotranspiration models should be used."*

**R2C26:** **Conclusions. "The general Budyko curve behavior can and should be utilized as a global constraint". The "should" seems a bit odd as there will be many instances in which there will be better/more data available than the Budyko curve to constrain models.**

AR: As is described in the remainder of the quoted sentence, we used the word "should" since any valid process-based evapotranspiration model must be able to reproduce the general Budyko curve behaviour when applied to multiple catchments across a range of climates since that behaviour is what is observed in nature. We agree that in many cases (e.g., for individual catchments) there will be better/more data to constrain evapotranspiration models (e.g., evapotranspiration from eddy covariance). However, it is still important to test that a specific model's structure is able to produce the general Budyko curve behavior when applied across a wide range of climates. To reflect this concept better in the manuscript, we revised a sentence to (**P28:L15-19**):

*"Additionally, to be a valid representation of catchment evapotranspiration, process-based models need to able to reproduce the empirically established, nonparametric Budyko curve behavior when applied to multiple catchments across a range of climates. Thus, the general Budyko curve behavior can serve as a global constraint (i.e., calibration or validation) in the application of such models, e.g., Greve et al. (2020)."*

**Dr. Randall Donohue, Short Comment 1 ()**

We thank the reviewer (Dr. Randall Donohue) for his time and helpful comments, which we have attempted to address and incorporate in manuscript revisions. Below are explanations of our responses to the reviewer's comments with all line numbers corresponding to the revised manuscript unless otherwise noted. The following notation is used to refer to the numbering of reviewer comments, pages, and line numbers: RXCY = Reviewer X Comment Y; SC = Short Comment; AR = Author Response; PX:LY = Page X Line Y.

**SCC1:** **I have eagerly read this manuscript and am welcoming of a new perspective on the Budyko curve. While the theoretical understanding of the curve has been continuing to grow, my perspective is that actual improvements to the performance of the model in hydrological applications stalled a few decades ago (my own work included!). It would be great to kick-start that process again. So I am glad to read a critical appraisal of the use of the Budyko curves.**

AR: We thank the reviewer for his interest in our manuscript and appreciate the kind words.

**SCC2: I have one comment to make on this manuscript, which relates to the claim that catchments in reality don't follow Budyko-like curves. The authors show this using catchment data from the US and the UK (Figure 2). The data show that the catchments do generally follow a curve when using the long-term averages but not when using time-series (looking at a catchment through time). This is an expected result due to the underlying assumption in the framework that precipitation is the \*only\* supply of water. This is often interpreted in terms of catchments needing to be in steady state.**

**Given this inherent model assumption, one should expect time-series catchment data NOT to follow a single curve. Hence, when the authors say that**

**"...we can conclude that individual catchments do not generally or consistently follow Budyko curve trajectories as posited by the catchment trajectory conjecture..." (page 17 line 8)**

**what the authors have (re)discovered is what happens when one violates a key model assumption.**

**Hence, their assertion that**

**"...this conjecture in hydrological analyses (e.g., precipitation partitioning sensitivity and causal attribution to anthropogenic and climatic impacts) will likely introduce significant errors and may lead to spurious conclusions."**

**seems to be difficult to support from their empirical test but also seems unfair both to the model itself and to those in the community who apply the model in accordance to its inherent limitations.**

> AR: First, we wish to make it clear that our intention is not to be unfair to the Budyko framework, nor to those who apply it. Our conclusions about the fundamental limitations of the parametric Budyko equations emerged from a genuine interest in the framework and a careful study of the catchment hydrology literature while attempting to improve the biophysical understanding of the catchment-specific parameters. After realizing the non-transferability and under-determined nature of the parametric Budyko equations in our own applications, we decided it would be beneficial to bring these issues to the larger catchment hydrology community, particularly given the large number of recent papers using the parametric framework. We have attempted to convey a message of recontextualization of prior results in the manuscript (e.g., **P3:L120-24** and **P23:L11-14**). However, we can likely improve our representation of this message, as also suggested by **Reviewer 1.** We addressed this by explicitly illustrating how the intent and efforts of prior work can be maintained if an appropriate interpretation of the parametric Budyko framework is applied (see edits to the manuscript in our responses to **R1C3** and **R1C16**).
>
> Next, we thank the reviewer for his accurate representation of the results from our empirical test of the catchment trajectory conjecture. Specifically, we found that the long-term behavior of hundreds of US and UK reference catchments generally follow the non-parametric Budyko curve when taken as an ensemble, however the behavior of individual catchments over time do not follow the conjectured parametric Budyko curves. The reviewer suggests, however, that time-series catchment data should not necessarily follow a single explicit curve, since such data violate the underlying steady state assumption of the Budyko approach. He therefore suggests that our conclusions about the catchment trajectory conjecture are not supported by our empirical test.
>
> However, the reviewer's interpretation neglects two central elements of empirical test methodology. Specifically, we used only reference catchments (i.e., stable catchment characteristics) and accounted for potential non-steady state storage impacts by testing

catchments' actual trajectories through Budyko space for essentially all possible temporal averaging windows (see further details in our response to **SCC3**). We thus believe that we fairly tested the important assumptions and interpretations of the parametric Budyko framework—and found them to be unsupported for individual catchments. While the ensemble behavior of many catchments generally follows the non-parametric Budyko curves, our evidence suggests the trajectories of individual catchments are not specific parametric curves.

To be clear, we agree that not accounting for storage dynamics is a violation of the Budyko framework's underlying assumption. However, we note that the catchment trajectory conjecture and methods derived from it specifically suggest that the temporal evolution of a catchment (i.e., its time series) will follow a particular Budyko curve under changes in aridity index. For example, the two derived methods that the reviewer quotes from our manuscript, precipitation partitioning sensitivity and causal attribution to anthropogenic and climatic impacts, are fundamentally based on the idea of change in a catchment's Budyko space trajectory over time. Combined with **SCC3-4**, we thus conclude that the reviewer is suggesting that time series data will follow a particular Budyko curve, as long as the catchment properties remain unchanged and the storage dynamics are properly accounted for (or the assumption of steady state is not violated). We agree with this interpretation of required conditions (i.e., stable catchment characteristics and steady state storage) and contend that our empirical test methodology meets these requirements and therefore provides a robust assessment of the claims of the catchment trajectory conjecture.

**SCC3: Is it not the case that only time-series (daily, weekly, yearly etc) hydrological data that have water storage change effects accounted for can be used to test, empirically, whether the catchment specific parameter is temporally constant?**

AR: Yes, in order to quantify $\bar{E}$ accurately for a catchment (which is required to test if the catchment-specific parameter is temporally constant), the remaining components of the water balance must be known since, $\bar{E} = \bar{P} - \bar{Q} - \overline{\Delta S}$. While the 728 references catchments used in our empirical analysis had daily time series of $P$ and $Q$ (which can be averaged over the time interval of interest to obtain $\bar{P}$ and $\bar{Q}$), they lacked estimates of $\Delta S$, as is the case for the vast majority of available catchment hydrology data. We recognized this potential limitation in our analysis and addressed it explicitly by testing the catchment trajectory conjecture for all possible relevant realizations of each catchment's actual trajectory through Budyko space (**P12:L7-19**). Specifically, we computed time-varying $\bar{P}$, $\overline{E_0}$, and $\bar{E}$ by applying moving-average window sizes ranging in annual steps from 1 year to the full length of record.

It should be expected that above some threshold averaging window size (e.g., 10-, 20-, 30-year average window), changes in catchment storage average to zero ($\overline{\Delta S} \sim 0$). This has been shown to be the case for catchments across Earth (Han et al., 2020), with 71% of catchments reaching steady state with a 10-year averaging window and 94% reaching steady state with a 30-year averaging window. However, even if this threshold behavior does not apply universally, it should be expected that, for some averaging windows and some catchments, steady state conditions would be present (i.e., $\overline{\Delta S} \sim 0$). In either case (threshold behavior or not), testing all the relevant averaging windows for all catchments allows for a robust test of the catchment trajectory conjecture. In the case of threshold steady state behavior, if the catchment trajectory conjecture is correct, actual and expected Budyko space trajectories for a catchment would be consistently and statistically indistinguishable once the averaging window reaches a sufficient length (**P12:L20-27**). Statistically, this means that the frequency at which actual and expected Budyko space trajectories were found to be statistically indistinguishable would be higher than expected at

random (i.e., more than 5% of all actual vs. expected trajectories would be statistically indistinguishable at a significance level of 0.05). An elevated frequency of statistical similarity would also occur if catchments were only rarely in steady state. The reason for this elevation is that the number of statistically similar trajectories from averaging windows where $\overline{\Delta S} \sim 0$ would be in addition to the number of statistically similar trajectories expected from random chance. Critically, the results of our empirical test presented in the main text show that the catchment trajectory conjecture is not supported. Out of the 24,501 actual trajectory realizations, 23,231 (95%) were found to have consistent differences (p-value < 0.05) from their "expected" trajectories, while only 1270 (5%) were found to be statistically indistinguishable. This proportionality is exactly what would be expected due to random chance.

We emphasize that our methodology directly addresses the (very common) lack of knowledge of catchment storage dynamics that the reviewer points out, and it thus provides a robust test of the catchment trajectory conjecture. However, we believe we can be clearer on this point in the manuscript, and so we made the following revisions:

> 1) Changed a section of text in Section 3.1.2 to (**P12:L7-25**):
>
> *"For a given reference catchment, estimates of $\bar{P}$ and $\overline{E_0}$ were obtained from daily records of $P$ and $E_0$, while estimates of $\bar{E}$ were calculated from the catchment water balance, $\bar{E} = \bar{P} - \bar{Q}$, which assumes impacts from storage dynamics are negligible ($\overline{\Delta S} \approx 0$). Since $\bar{P}$, $\overline{E_0}$ and $\bar{E}$ represent temporal averages, and we were also interested in temporal trajectories of those magnitudes, we computed time series of moving averages for each of the three variables, similar to the method employed by van der Velde et al. (2014). The temporal averaging window for which $\overline{\Delta S} \approx 0$ is typically unknown, however it has been shown to exhibit a threshold behavior (i.e., above a certain averaging window size $\overline{\Delta S}$ is consistently negligible) (Han et al., 2020). The threshold averaging window size can vary between catchments, but approximately 71% of global catchments reach the threshold with an averaging window of 10 years, and 94% of catchments reach the threshold with an averaging window of 30 years (Han et al., 2020). To address the uncertainty in the threshold averaging window size, we computed different "realizations" of the actual trajectories in terms of $\frac{\overline{E_0}}{\bar{P}}$ and $\frac{\bar{E}}{\bar{P}}$ for each catchment for all possible integer-year averaging windows in annual steps from one year to the full length of record. The "conjectured" Budyko curve of Eq. (5) was fitted by adjusting the value of $n$ using the full length of record in each catchment.*
>
> *The conjecture was tested for each reference catchment by comparing all realizations of actual trajectories to the conjectured Budyko curve trajectory using the non-parametric sign test (Holander and Wolfe, 1973). This is a distribution-free test for consistent over- or under-estimation between paired observations (see also Supplemental Information Sect. S2). If the catchment trajectory conjecture is correct, then the frequency at which actual and expected Budyko space trajectories are found to be statistically indistinguishable will be higher than what is expected due to random chance (see also Supplemental Information Sect. S2)."*
>
> 2) Added a new section to the supplementary information (**Section S2.5**) describing how we controlled for potential catchment storage dynamics.

**SCC4: Might it be that water storage provides the link between the analytical meaning of the parameter and the physical understanding of how individual catchments have different E for the same P and Ep?**

**That is, when a stored water term is included as a water supply term alongside P, then it introduces a catchment-specific and time-dependent term into the model fundamentals and into your analytical analysis?**

> AR: We agree that it might be possible that explicitly including storage changes in the water supply term (i.e., $\bar{E} = [\bar{P} - \overline{\Delta S}] - \bar{Q}$) would show that reference catchments' trajectories follow a particular parameter Budyko curve with a temporally constant catchment specific parameter (a "storage-dependent catchment trajectory conjecture"). However, to our knowledge, this hypothesis has not been explicitly tested within the literature. Additionally, the results from our empirical test (which implicitly incorporates storage dynamics) do not support this idea (see our response to **SCC3** above). Therefore, while it might be possible, the best currently available evidence does not appear to support such as hypothesis.

> Furthermore, if the storage-dependent catchment trajectory conjecture is indeed correct, it can only be correct for one of the formulations of the parametric Budyko equations, since the various functional forms produce contradicting curves/trajectories when the catchment-specific parameter is held constant (see **Sections 3.2.2** and **4.2.2** of the main text).